# BEYOND REASONING GAINS: MITIGATING GENERAL CAPABILITIES FORGETTING IN LARGE REASONING MODELS

## ABSTRACT

Reinforcement learning with verifiable rewards (RLVR) has delivered impressive gains in mathematical and multimodal reasoning and has become a standard post-training paradigm for contemporary language and vision-language models. However, the RLVR recipe introduces a significant risk of capability regression, where models forget foundational skills after prolonged training without employing regularization strategies. We empirically confirm this concern, observing that open-source reasoning models suffer performance degradation on core capabilities such as perception and faithfulness. While imposing regularization terms like KL divergence can help prevent deviation from the base model, these terms are calculated on the current task, thus they do not guarantee broader knowledge. Meanwhile, commonly used experience replay across heterogeneous domains makes it nontrivial to decide how much training focus each objective should receive. To address this, we propose a replay strategy with dynamic objective reweighting for general knowledge preservation. Our reweighting mechanism adapts in an online manner using short-horizon signals of convergence and instability, shifting the post-training focus away from saturated objectives and toward underperforming or volatile ones. Our method is end-to-end and readily applicable to existing RLVR pipelines without training additional models or heavy tuning. Extensive experiments on benchmarks based on Qwen2.5-VL-3B and Qwen2.5-VL-7B demonstrate the effectiveness of our method, which not only preserves general capabilities but also improves reasoning by enabling more flexible trade-offs among in-task rewards.

## 1 INTRODUCTION

Large Language Models (LLMs) and Vision-Language Models (VLMs) have demonstrated remarkable general-purpose capabilities (Achiam et al., 2023; Yang et al., 2023), yet strengthening their proficiency in complex reasoning remains a key frontier of research. Reinforcement Learning with Verifiable Rewards (RLVR) (Shao et al., 2024), an extension of Reinforcement Learning from Human Feedback (RLHF) (Ziegler et al., 2019; Ouyang et al., 2022), has emerged as a powerful paradigm for this purpose. By providing explicit reward signals such as exact-match correctness, format adherence or brevity in final answers, RLVR has been applied to instruction following, STEM problem solving, code generation and logical reasoning models (Lightman et al., 2023b; Peng et al., 2025), resulting in large performance gains on benchmark scores, leading to headlines that language models can "learn to reason" (Guo & DeepSeek-AI, 2025).

Despite strong headline gains, RLVR exhibits recurring failure modes, prompting questions about whether current pipelines genuinely expand reasoning abilities (Shojaee et al., 2025). For example, exploration and diversity collapse occur when on-policy finetuning overly narrows the policy distribution—raising Pass@1 but reducing Pass@k and solution-path diversity (Yue et al., 2025; Dang et al., 2025). Likewise, outcome-only rewards introduce sparse credit assignment and instability, and not every task is naturally cast as a reinforcement-learning problem (e.g., translation, summarization, or captioning). In addition, strict answer formats and format-sensitive graders may conflate genuine reasoning improvements with mere format compliance, even introducing evaluation artifacts (Petrov et al., 2025). Recent studies report that many RL-trained models even underperform the base model in standardized evaluation, where formatting-reward only baseline degrades the original performance

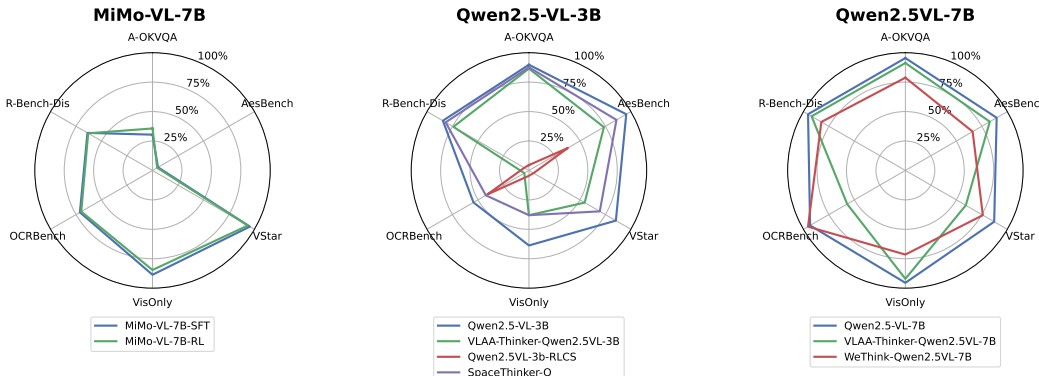

Figure 1: **General capabilities comparison of base VLMs (blue) and their reasoning-tuned variants (green/purple/red) on six representative, non-reasoning benchmarks (higher is better)**: A-OKVQA (knowledge-based VQA), AesBench (Huang et al., 2024b) (image aesthetics) , VStar (Wu & Xie, 2023) (spatio-temporal reasoning), VisOnly (Kamoi et al., 2025) (vision-only recognition aggregate) , OCRBench (Liu et al., 2024b) (text recognition), and R-Bench-Dis (Li et al., 2025b) (distribution-shift robustness). Across both Qwen2.5-VL families, reasoning-finetuned models generally underperform their base models on perception and robustness tasks, whereas MiMo-VL-7B-RL remains close to its SFT baseline.

more severely (Prabhudesai et al., 2025). The format reward model might be underoptimized or optimizing the format reward causes forgetting of math capabilities (Chandak et al., 2025).

Another critical yet under-explored issue in RLVR is that optimizing for a narrow set of targeted, verifiable rewards can lead to regression in general capabilities acquired during pretraining. Although models become proficient in following formatting requirements and solving reasoning tasks, they simultaneously exhibit increased hallucinations (Jaech et al., 2024; Yao et al., 2025b) and are more vulnerable to jailbreak attacks (Lou et al., 2025; Yao et al., 2025a). These results suggest that reasoning-oriented post-training can improve reasoning but at the cost of trading off non-target competencies (e.g., perception, safety, factual grounding), especially when prolonging the RL training without explicitly applying regularization (Liu et al., 2025a).

To examine the forgetting issue of reasoning-focused finetuning, we begin by probing the general abilities of open-source reasoning models beyond their target reasoning tasks. As shown in Fig. 1, models finetuned for chain-of-thought or RL reasoning frequently lag behind their base counterparts on perception and robustness. For example, we observe the consistent drop on VisOnlyQA across both Qwen2.5-VL families, while MiMo-VL-7B-RL performs competitively against its base model on those non-reasoning tasks. We hypothesize this is due to their special finetuning strategy, where they employ mixed on-policy reinforcement learning that tries to maximize the model capacity on multiple axes beyond math and reasoning, according to the MiMo-VL-7B technical report (Team et al., 2025). However, the detailed framework and the sampling or reweighting are not disclosed. These patterns support our central claim: optimizing for reasoning rewards can erode non-reasoning capabilities, motivating a continual learning method to preserve general skills during reasoning-oriented post-training.

Our initial experiments with Qwen2.5-VL-7B show that training solely on reasoning rewards degrades performance on general capabilities, for example, by 7% on the perception task, according to Figure 2. To address this, we propose to gather general-capability data and integrate it into RLVR via an online

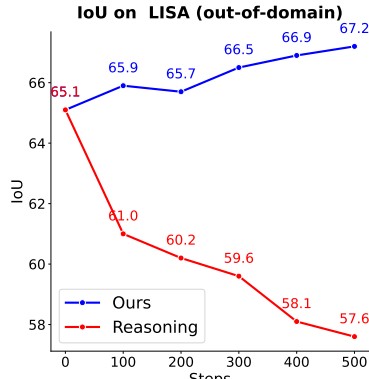

Figure 2: **Performance comparison between the model finetuned solely on math data and with general data replay**. Our model not only preserves the base model performance but also improves it (2%) while the reasoning model quickly falls behind the base model after 100 iterations.

weighting mechanism. However, due to cross-domain heterogeneity, it is nontrivial to decide how much weight to assign to each loss term or reward. We then measure how different reward signals

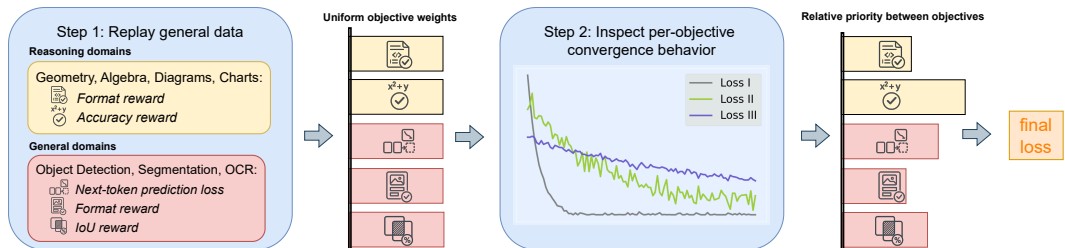

Figure 3: **Overview of our proposed method.** Along with the target reasoning task, we sample data from general domains in order to maintain that knowledge during finetuning. Initially, the objectives of interest are reweighted uniformly to optimize the main model. After a few iterations, we record the convergence behavior of individual objectives. Based on this, we adjust the focus to prevent the dominance of any objectives and put less weight on saturated ones.

evolve during RLVR training in Section 4 and find that some rewards converge up to three times faster than others, which should therefore not be emphasized later in training when the model is already able to obtain the corresponding skills.

Motivated by the above observations, we propose a principled replay-based training strategy that mixes general data back into the RL objective, then dynamically reweights objectives based on their convergence rate and instability. As shown in Figure 3, our proposed method computes the relative priority between objectives of interest by inspecting their convergence behaviors to reweight them in the final loss. Experiments demonstrate that our method not only preserves general capabilities but also improves reasoning performance by allowing flexible trade-offs among reward types. In summary, our contributions are depicted as follows:

- We systematically re-evaluate open-source reasoning models and show that reasoning-focused finetuning consistently regresses general capabilities. This motivates replaying general-capability data during RLVR to preserve pretraining knowledge. We further show that objectives exhibit distinct convergence behaviors, making commonly used, manually tuned reweighting schemes suboptimal in such scenarios.

- We introduce a plug-in scheduler that replays general-capability data during RLVR and dynamically reweights both general and main task objectives. Our proposed method naturally down-weights saturated format signals and refocuses capacity on harder, high-variance objectives. The method is end-to-end, magnitude-agnostic, requires no auxiliary models, and drops into RLVR pipelines without overhead.

- In our experiments, our method preserves or improves general capabilities while matching or exceeding the reasoning performance of reasoning-only finetuning. It consistently outperforms strong continual-finetuning baselines and is competitive with specialized open models while utilizing smaller compute. Empirically, we also find that replaying general data yields shorter, more concise rationale while not compromising reasoning ability.

## 2 RELATED WORK

**Foundation models and post-training.** Large transformer models pretrained on broad corpora serve as general-purpose backbones with strong abilities and wide transfer across domains (Brown et al., 2020; Touvron et al., 2023). Post-training adapts these backbones to downstream tasks via (i) supervised finetuning, from early ULMFiT (Howard & Ruder, 2018) to instruction tuning in FLAN (Wei et al., 2021) or Flan-T5 (Chung et al., 2024); (ii) reinforcement learning from human or AI feedback, typically combining preference modeling with policy optimization; and (iii) direct preference optimization objectives that bypass explicit reward models. For reasoning, reinforcement learning with verifiable rewards has become a common recipe: verifiers or rule-based checkers score final solutions in math and related domains, often within PPO-style pipelines (Guo & DeepSeek-AI, 2025; Liu et al., 2025b). Process-based neural reward models provide supervision for intermediate progress (Setlur et al., 2025) rather than only the final output. However, PRMs can induce reward hacking: agents learn to exploit the appearance of correct process rather than achieving the intended outcome (Wang et al., 2025a; Shao et al., 2024).

**Catastrophic forgetting in continual and post-training.** Catastrophic forgetting describes performance regressions on previously acquired skills when adapting to new data (McCloskey & Cohen, 1989; French, 1999). Early work in this vein introduced regularization-based mitigations such as Elastic Weight Consolidation (Kirkpatrick et al., 2017), Synaptic Intelligence (Zenke et al., 2017), and Memory-Aware Synapses (Aljundi et al., 2018) that prevent excessive change on important parameters. Functional approaches like Learning without Forgetting (Li & Hoiem, 2016) constrain outputs via distillation (Hinton et al., 2015), and replay via small episodic memories (Rebuffi et al., 2017; Lopez-Paz & Ranzato, 2017; Rolnick et al., 2019; Buzzega et al., 2020) are consistently strong baselines across settings. This problem is not unique to older neural networks, recent studies show that large language models also forget under sequential post-training. For instance, continual instruction tuning degrades domain knowledge, reasoning, and reading-comprehension performance unless replay or modularization is introduced (Luo et al., 2023; He et al., 2023; Wang et al., 2024; Huang et al., 2024a; Schen et al., 2024).

**Mitigations in post-training practice.** Practical pipelines increasingly combine RL with regularization terms or replay knowledge to regularize the policy, preventing it from shifting too far from the original base model. Along with the standard KL-regularization approaches, InstructGPT (Ouyang et al., 2022) interleaves the pretraining gradients with RLHF updates to reduce drift relative to KL-only regularization (Zheng et al., 2023). Concurrent works (Zhang et al., 2025; Fu et al., 2025) integrate verified rollouts to stabilize learning or penalize the discrepancy on augmented training data (Wang et al., 2025c). Yet, those methods do not guarantee the performance preservation on non-target domains. Other approaches tackle this by incorporating mixed, verifiable reward suites (Team et al., 2025) or introducing reflection or re-attention mechanisms under RL objectives (Chu et al., 2025). In addition, recent reasoning-focused RL pipelines often reduce or remove KL to encourage exploration (Hu et al., 2025a; Hao et al., 2025), potentially exacerbating forgetting.

## 3 BACKGROUND

**Supervised finetuning (SFT).** Let an LLM with parameters $\theta$ induce a conditional policy $\pi_\theta(\cdot \mid x)$ over responses $y$ to a prompt $x$. SFT optimizes the negative log-likelihood on instruction–response pairs $\mathcal{D} = \{(x^{(i)}, y^{(i)})\}_{i=1}^N$:

$$\mathcal{L}_{\text{SFT}}(\theta) = -\sum_{i=1}^{N} \log \pi_\theta\big(y^{(i)} \mid x^{(i)}\big).$$

SFT has been central to transferring general-purpose LMs to downstream instruction following and broad zero-shot generalization and it typically provides the initialization for subsequent preference- or reward-based alignment.

**RL-based post-training.** Reinforcement learning from human feedback (RLHF) fits a reward model $r_\phi(x, y)$ from pairwise human preferences (Ziegler et al., 2019; Rafailov et al., 2023; Lambert, 2025), commonly using a Bradley–Terry likelihood (Bradley & Terry, 1952), and then maximizes reward while regularizing toward the pretrained reference $\pi_{\text{ref}}$ (often with a KL penalty) via policy optimization such as PPO (Schulman et al., 2017):

$$\max_\theta \ \mathbb{E}_{x \sim \mu, \ y \sim \pi_\theta(\cdot \mid x)}\Big[r_\phi(x, y)\Big] \ - \ \beta \, \mathbb{E}_{x \sim \mu}\Big[D_{\text{KL}}\big(\pi_\theta(\cdot \mid x) \,\|\, \pi_{\text{ref}}(\cdot \mid x)\big)\Big].$$

This pipeline improves helpfulness/harmlessness while retaining base-model competence; see early LM-preference work and InstructGPT for canonical formulations, and PPO for the underlying stable policy-gradient updates (Stiennon et al., 2020). Preference modeling itself is often grounded in the classical Bradley–Terry model for pairwise comparisons (Rafailov et al., 2023).

**Reinforcement learning with verifiable rewards (RLVR).** In settings with programmatic or automatic verifiers (e.g., exact-match answers, execution-based checks, or constraint checkers), RLVR replaces learned human-preference rewards with verifiable signals $r(x, y) \in [0, 1]$. This reduces labeler noise and can better target reasoning fidelity by rewarding demonstrably correct steps or outcomes, while retaining the same KL-regularized RL form (Lightman et al., 2023b).

**Group Relative Policy Optimization.** GRPO is a PPO-style algorithm tailored for LLM reasoning that forgoes a learned value/critic and instead computes advantages from group-normalized sequence rewards. For each prompt $x$, sample a group of $G$ rollouts $O = \{o_i\}_{i=1}^{G}$ from a frozen rollout policy $\pi_{\theta_{\text{old}}}$. Let $R_i$ be the verifiable sequence-level reward and define the group-normalized advantage $\widehat{A}_i = \big(R_i - \text{mean}(R)\big)/\text{std}(R)$. With token-wise importance ratio $r_{i,t}(\theta) = \frac{\pi_\theta(o_{i,t}|x,o_{i,<t})}{\pi_{\theta_{\text{old}}}(o_{i,t}|x,o_{i,<t})}$, GRPO maximizes the clipped surrogate plus KL regularization:

$$
\mathcal{J}_{\text{GRPO}}(\theta) = \mathbb{E}\left[ \frac{1}{G}\sum_{i=1}^{G} \frac{1}{|o_i|} \sum_{t=1}^{|o_i|} \min\Big\{ r_{i,t}(\theta)\widehat{A}_i,\ \text{clip}\big(r_{i,t}(\theta), 1-\epsilon, 1+\epsilon\big)\widehat{A}_i \Big\} \ - \ \beta\, D_{\text{KL}}\big(\pi_\theta \,\|\, \pi_{\text{ref}}\big) \right].
$$

By normalizing across a group of responses per prompt, GRPO stabilizes updates without a critic, which is preferable for long chain-of-thought rewards that are sparse and verifier-based. Empirically, GRPO boosts mathematical-reasoning performance in many open models (Shao et al., 2024).

**General ability degradation.** Unlike traditional continual learning studies, modern post-training pipelines must jointly consider gains in reasoning and retention of inherent general abilities (e.g., perception, grounding, instruction following, safety). Let $\mathcal{G} = \{G_1, \ldots, G_M\}$ denote a suite of general-ability tasks and $\mathcal{R} = \{R_1, \ldots, R_L\}$ a suite of reasoning datasets. Earlier efforts (Ziegler et al., 2019; Nakano et al., 2021; Korbak et al., 2022) in mitigating this drift adopt KL penalties to let the policy model move toward higher rewards while penalizing large shifts from the pretrained model. However, this regularization term is imposed on the target domain data and does not guarantee retention of arbitrary non-target skills.

# 4 OUR PROPOSED METHOD

We address forgetting in RLVR by (i) replaying general-capability data alongside reasoning data, and (ii) dynamically reweighting objectives online using the local estimation of progress and instability for individual objectives. Below, we present how our proposed method governs loss coefficients and shifts the optimization away from saturated objectives toward underperforming or volatile ones—without changing the underlying RL algorithm.

**Setting.** In the context of supervised learning, let $\mathcal{D} = \{\mathcal{D}_n\}_{n=1}^{N}$ be $N$ domains and $\ell_{n,k}^{(t)}(\theta)$ the mini-batch loss of objective $k \in \{1, \ldots, K\}$ on domain $n$ at iteration $t$ for parameters $\theta$. Note that $K \geq N$ as there are some tasks using more than one reward or objective. The model parameter $\theta$ is thus optimized by minimizing the average loss across objectives:

$$
L_k^{(t)} \ = \ \frac{1}{N} \sum_{n=1}^{N} \ell_{n,k}^{(t)}(\theta).
$$

Our framework acts on $\{L_k^{(t)}\}_{k=1}^{K}$ regardless of whether each $L_k$ arises from an RL reward surrogate or a supervised learning loss term.

**Per-objective rate and the stability of convergence.** Due to the unstable nature of RL training, we can not rely solely on the per-step objective/reward value to compute the reweighting coefficients. Instead, for each objective $k$, we measure the *convergence rate* over a sliding window of length $2 \times W$ by computing the current window average and the previous window average by:

$$
\underbrace{\mu_k^{(t)} = \frac{1}{W} \sum_{s=t-W+1}^{t} L_k^{(s)},}_{\text{estimated current loss value}} \qquad\qquad \underbrace{\tilde{\mu}_k^{(t)} = \frac{1}{W} \sum_{s=t-2W+1}^{t-W} L_k^{(s)}}_{\text{estimated old loss value}}
$$

and the *instability* (coefficient of variation) in the same window:

$$
\sigma_k^{(t)} = \sqrt{\frac{1}{2W-1} \sum_{s=t-2W+1}^{t} \big(L_k^{(s)} - \mu_k^{(t)}\big)^2}.
$$

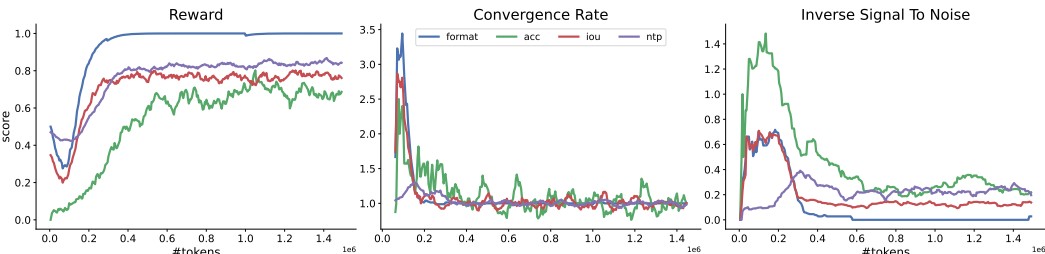

Figure 4: **Different rewards have different behaviors of convergence.** While the format reward is easy to optimize and obtains the highest rate of convergence, it quickly saturates and thus yields low convergence rate ($c \sim 1$) and instability ($i \sim 0$) after 50 steps. By contrast, the reasoning accuracy fluctuates the most, thereby steering the optimization toward the corresponding objective. IoU and ntp indicate the IoU reward and the next-token-prediction accuracy during training. The result is obtained on the first setting in our experiment section below.

Based on these measures, we form two signals (i) the $\boxed{convergence\ rate\ c_k^{(t)} = \tilde{\mu}_k^{(t)}/\mu_k^{(t)}}$ captures how fast the loss is improved, while (ii) the $\boxed{inverse\ signal\text{-}to\text{-}noise\ ratio\ i_k^{(t)} = \sigma_k^{(t)}/(\mu_k^{(t)} + \tilde{\mu}_k^{(t)})}$ captures loss instability. Intuitively, $c_k^{(t)} > 1$ indicates recent improvement (loss dropping relative to the previous window), while $c_k^{(t)} \approx 1$ signals saturation. The term $i_k^{(t)}$ is larger when the objective is noisy/unstable.

**Relative priority between domains.** We convert these signals into normalized coefficients via a temperature-controlled softmax. With the temperature $T > 0$, we define the priority of $k$-objectives as $s_k^{(t)}$ and compute the coefficients $\lambda$ for reweighting objectives.

$$s_k^{(t)} = c_k^{(t)} + i_k^{(t)}, \qquad \lambda_k^{(t)} = \frac{K \exp\!\big(s_k^{(t)}/T\big)}{\sum_{i=1}^{K} \exp\!\big(s_i^{(t)}/T\big)}. \tag{1}$$

The prefactor $K$ preserves average scale so that $\frac{1}{K}\sum_k \lambda_k^{(t)} = 1$. Lower $T$ sharpens priorities while higher $T$ approaches uniform mixing. We set $T = 5$ by default in our experiments.

**Overall training objective.** At step $t$, we minimize the following weighted objective:

$$\mathcal{L}^{(t)}(\theta) = \frac{1}{K}\sum_{k=1}^{K} \lambda_k^{(t)}\, L_k^{(t)}.$$

Optimizing $\theta$ with $\nabla_\theta \mathcal{L}^{(t)}$ steers learning towards objectives that are *both* slow-to-converge (high $c_k$) *and* fluctuated (high $i_k$), while leaving well-learned, stable objectives with lower weight. The scheme reduces to the standard equal weighting as $T \to \infty$ (i.e., uniformly sample from all domains without loss reweighting).

Figure 4 illustrates the insight behind our proposed method, where we finetune the model on a reasoning dataset (tracked by accuracy) while replaying perception data (IoU) and an SFT dataset for generality. Earlier in training, the format signal is easy to optimize and saturates quickly, so its $c$ falls toward 1 and $i$ toward 0, reducing its priority. After 100 steps, many signals plateau (convergence rate $c \approx 1$) while they still differ in the level of stability. Among them, the reasoning reward remains fluctuating the most ($\sim 0.3$, yielding higher $i$ and thus higher $\lambda$. At this point, the model has learned to answer according to the predefined template; thus the corresponding signal-to-noise ratio for formatting reward is $\approx 0$. This motivates combining *both* progress ($c$) and instability ($i$) Equation 1 as they complement each other. Finetuning the trade-off between those two terms offers finer-grained control and potentially improves the performance. However, we simply take their unweighted sum $s = c + i$ for simplicity, which performs consistently well in our experiments.

Following the illustrative setup of Navon et al. (2022), we consider a synthetic two-task problem with a shared parameter vector and two scalar objectives. The corresponding Pareto front can be computed analytically and is shown in gray in Figure 5. We benchmark our method against established loss-magnitude-based methods (first row) and gradient-based multi-task learning methods (second row).

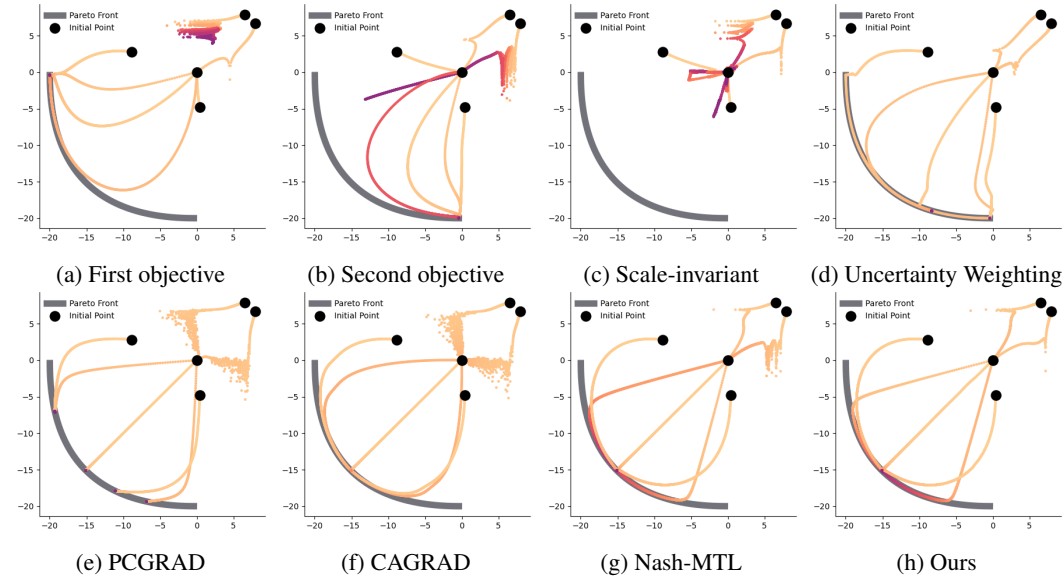

(a) First objective    (b) Second objective    (c) Scale-invariant    (d) Uncertainty Weighting

(e) PCGRAD        (f) CAGRAD        (g) Nash-MTL        (h) Ours

Figure 5: A modified illustrative two-task example from (Navon et al., 2022) to show the convergence of comparative methods from different initialization points (black dots ●). Each optimization trajectory is colored from yellow to red. The bold gray line represents the Pareto front. Overall, while gradient-based MTL methods such as CAGRAD and NashMTL do not depend on the initial solutions, they suffer from slow convergence and larger per-step computational cost compared to ours.

To mimic the unstable nature of RL training, we inject noise into the first objective, which induces substantial fluctuations for competing methods, whereas our approach remains stable and closely tracks the Pareto front. Additional runtime comparisons in the appendix highlight that our method also achieves favorable wall-clock efficiency, a crucial advantage for large-scale RL training.

## 5 EXPERIMENTS

In this section, we demonstrate how our proposed method can be used to replay general capabilities data during RLVR under two complementary experimental settings. First, we evaluate under the **RLVR-Only Setting** (Liang et al., 2025), which serves as our most comparable baseline where they mix diverse datasets in RLVR. Second, since MoDoMoDo enforces binary (0–1) rewards across all domains and thus cannot directly handle non-RL tasks, we further extend our evaluation to a **Hybrid Setting** that integrates RL-style and supervised (non-RL) objectives, bringing it closer to the standard practices used in large-scale LLM training.

### 5.1 EXPERIMENTAL SETTINGS

To evaluate our proposed approach, we conduct our experiments using the two base models: Qwen2.5-VL-3B and Qwen2.5-VL-7B on two setups in different scales: **RLVR-Only Setting:** This smaller setup follows the experimental configuration of Liang et al. (2025), focusing on domain-specific RLVR, which could serve as the *upper-bound* baseline of static data mixture approaches. The Qwen2.5-VL-3B model is trained until data from a particular domain is exhausted. We train on 8 GPUs using data parallelism, with a per-device batch size of 2 and 4 rollouts per prompt. We evaluate this setup on six reasoning benchmarks: SAT (Ray et al., 2024a), ScienceQA (Lu et al., 2022a), MathVista (Lu et al., 2023), ChartQA (Masry et al., 2022), InfoVQA (Mathew et al., 2022), and MMMU (Yue et al., 2023). **Hybrid Setting:** To bridge RL and supervised training paradigms, we finetune Qwen2.5-VL-7B under a larger mixed objective regime that combines RLVR with SFT-style training. Specifically, we use ThinkLite-VL-70k (Wang et al., 2025b) while jointly replaying perception-oriented datasets such as RefCOCO (Kazemzadeh et al., 2014) and LLaVA-OneVision OCR (Li et al., 2024). Each model is trained for 500 steps on 8 GPUs using data parallelism, with per-device batch size 1, 2 gradient accumulation steps (effective batch size 16), and 4 rollouts per prompt (64 rollouts per optimizer step). For this larger hybrid setting, we evaluate our models on a

Table 1: **Benchmark results in RLVR-only setting.** We report the accuracy score over six benchmarks, in which the MoDoMoDo baseline is trained to maximize performance. Please note that in this benchmark only, we use the rule-based evaluator on the MathVista dataset instead of "gpt-3.5-turbo" to align with Liang et al. (2025).

| Model | SAT | ScienceQA | MathVista (mini) | ChartQA | InfoVQA | MMMU |
|---|---|---|---|---|---|---|
| *Open-source reasoning baselines* | | | | | | |
| VLAA-Thinker-3B | 49.38 | 14.63 | 30.4 | 45.84 | 30.81 | 32.22 |
| MM-R1-MGT-PerceReason | 50.83 | 34.21 | 33.4 | 44.88 | 61.42 | 40.22 |
| Ocean_R1_3B_Instruct | 59.49 | 68.72 | 38.7 | 54.00 | 38.02 | 40.89 |
| Qwen2.5VL-3b-RLCS | 24.12 | 21.32 | 17.2 | 3.32 | 10.86 | 27.11 |
| vision-grpo-qwen-2.5-vl-3b | 50.57 | 4.17 | 32.4 | 67.80 | 58.29 | 37.22 |
| Qwen2.5-VL-3B-Instruct-GRPO-deepmath | 34.70 | 45.27 | 32.3 | 70.24 | 49.75 | 39.11 |
| *Qwen2.5-VL-3B and our variants* | | | | | | |
| Base model | 43.98 | 6.20 | 23.6 | 43.88 | 32.02 | 38.67 |
| Uniform | 44.55 | 64.85 | 32.4 | 69.68 | 58.30 | 39.44 |
| MoDoMoDo | 49.95 | 65.74 | 32.2 | **70.40** | 59.88 | 39.11 |
| **Ours** | **55.19** | **71.59** | **33.2** | **70.40** | **60.78** | **42.44** |

broad suite of widely used VLM benchmarks spanning general multimodal understanding, visual reasoning, math-in-vision, OCR, and accessibility: LISA (Lai et al., 2024), MMMU-Pro (Yue et al., 2024), AI2D (Kembhavi et al., 2016), MathVista (Lu et al., 2023), MathVision (Liu et al., 2024c), MathVerse (Zhang et al., 2024b), MMBench (Liu et al., 2024a), VizWiz (Gurari et al., 2018), and OCRBench v2 (Fu et al., 2024). For the smaller setup, we also follow (Liang et al., 2025) and evaluate the model on SAT (Ray et al., 2024a), ScienceQA (Lu et al., 2022a), MathVista (Lu et al., 2023), ChartQA (Masry et al., 2022), InfoVQA (Mathew et al., 2022), and MMMU (Yue et al., 2023).

To isolate the effect of replay and dynamic reweighting and also for the ease of convenient implementation, we uniformly sample across data sources by default and reweight only the objectives of interest. Unless otherwise noted, we disable the reference-KL penalty to disentangle the effectiveness of regularization approaches (Li & Hoiem, 2017) and our replay mechanism. We also include the comparison against this approach in our list of established baselines for continual learning below:

- **Reasoning only:** We train solely on the target reasoning task with fixed reward weights (no replay). This is the most straightforward approach in continual learning.

- **Default:** General data is included during finetuning, with data across domains sampled in proportion to their source size. Losses are not reweighted.

- **Uniform:** Data are sampled uniformly across domains. Losses are not reweighted.

- **Coreset:** Replay a size-limited subset of general data (half the reasoning-data volume in our setup) to align with standard coreset-style replay methods (Rebuffi et al., 2017; Chaudhry et al., 2019).

- **LwF:** Data are sampled uniformly, and we set the KL regularization coefficient $\beta$ to 0.01. We refer to this method as LwF, as it shares a similar approach to Learning without Forgetting (Li & Hoiem, 2017).

For context, we also include some representative open-source vision language models specializing in reasoning derived from the corresponding base models in each experiment. Please note that we list them here for easier benchmarking and we are not aiming to outperform them, as those models often undergo many complicated training pipelines. Models are evaluated with LMMS-Eval (Zhang et al., 2024a).

## 5.2 EXPERIMENTAL RESULTS

According to Table 1, RL training on the reasoning domain helps improve the performance of the base model consistently on both reasoning and perception benchmarks. Especially on SCIENCEQA, RL lifts the performance of Qwen2.5-VL-3B from 6 to 60. On this benchmark, our proposed method even outperforms comparative open-source reasoning models. We consider MoDoMoDo as the *upper-bound* approach of static data mixture approaches due to (i) MoDoMoDo has access to the target tasks performance during finetuning, which requires rerunning the experiments if new target tasks

are introduced (ii) they train multiple proxy models of the same size as the baseline models to learn the test performance as a function of mixing ratio, which is computationally expensive, especially in the context of reinforcement learning. Even after selecting an "optimal" mixture, the method still depends on hand-tuned reward weights (e.g., doubling accuracy and IoU relative to formatting rewards). Those trade-off coefficients are also set differently in prior work without clarification, which limits the generality.

Table 2: **Benchmark results in large hybrid setting.** We report accuracy scores (higher is better) on nine perception and reasoning benchmarks. Rows above the break are open-source reasoning models with different backbones; the lower block compares variants finetuned from the same Qwen2.5-VL-7B base model. Bold = best; underline = second best within the Qwen2.5-VL-7B family.

| Model | LISA | MMMU-PRO | AI2D | MathVista | MathVision | MathVerse | MMBench | VizWiz | OCRBenchv2 |
|---|---|---|---|---|---|---|---|---|---|
| *Open-source reasoning baselines* | | | | | | | | | |
| VLAA-Thinker-7B | 63.14 | 26.30 | 75.45 | 63.90 | 11.18 | 29.87 | 75.95 | 47.57 | 40.23 |
| Vision-R1-7B | 47.30 | 26.76 | 0.00 | 61.80 | 18.75 | 23.32 | 69.46 | 53.12 | 24.63 |
| OpenVLThinker-7B | 42.73 | 21.79 | 59.94 | 59.10 | 5.59 | 19.26 | 71.53 | 52.89 | 28.30 |
| *Qwen2.5-VL-7B and our variants* | | | | | | | | | |
| Base model | 65.13 | 25.55 | 67.62 | 61.70 | 9.54 | 26.29 | 71.82 | 50.82 | 39.49 |
| LwF | 65.08 | 29.59 | 73.93 | 63.90 | 18.42 | 33.98 | 73.11 | 53.12 | 39.56 |
| Default | 66.80 | 31.39 | 75.32 | 63.40 | 21.05 | 34.75 | 73.54 | 57.05 | 37.60 |
| Uniform | 65.18 | 31.91 | 76.43 | 65.60 | 22.13 | 36.07 | 75.34 | 54.05 | 38.06 |
| Coreset | 64.82 | 31.91 | **79.92** | **66.90** | 23.36 | 37.58 | 78.09 | **63.76** | 35.49 |
| Reasoning-only | 57.58 | 33.87 | 74.97 | 65.50 | 24.87 | 40.74 | 77.84 | 62.45 | 38.55 |
| **Ours** | **67.24** | **34.15** | 78.21 | 66.70 | **25.11** | **40.83** | **78.52** | 61.97 | **39.72** |

Table 2 reports the performance of models finetuned from Qwen2.5-VL-7B across different benchmarks in the hybrid setting, showcasing a more general scenario than RLVR-only training. Overall, our proposed method achieves the best or runner-up performance across different datasets, except for VizWiz where we still improve the base model performance by more than $10\%$. Compared with the base model, naively finetuning on the reasoning domain causes significant forgetting, especially on tasks not requiring extensive thinking. For example, finetuning on ThinkLite-VL-70k reduces the segmentation ability from 65.13 to 57.58 on LISA. Meanwhile, replaying the old data helps preserve the performance for all baselines on this dataset ($\approx 64$). Compared with uniform sampling, LwF achieves similar scene understanding performance while obtaining lower scores on the reasoning benchmark (e.g., 29.59 vs 31.91 on MMMU-PRO). Similar behaviors are observed in (Wang et al., 2025c; Hu et al., 2025a), where they remove this term for more plasticity. Among all baselines, our method obtains the highest segmentation score, boosting the performance of the base model by more than $2\%$. This improvement highlights the impact of loss reweighting over the uniform baseline.

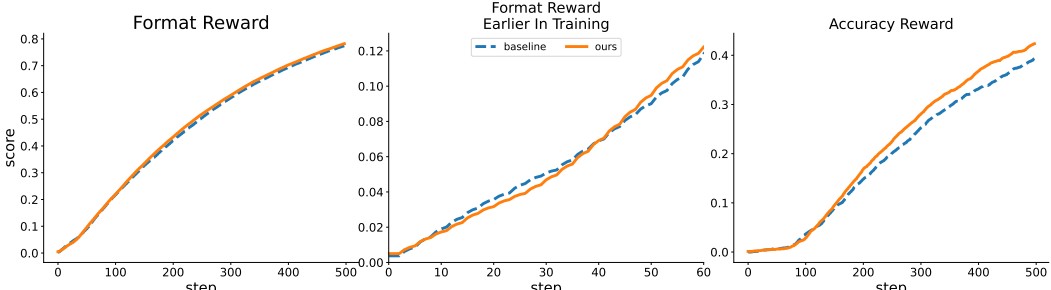

Figure 6: **Evolution of format and accuracy rewards on the reasoning domain during training:** Training curves for the format reward over full training (left), an early-training zoom (middle), and the accuracy reward (right). While the uniform baseline is better in maximizing the format reward, it falls behind our proposed method later in terms of accuracy, as we prioritize correct solutions over formatting once the model can follow the predefined template. Curves are smoothed with an exponential moving average for readability.

## 5.3 ABLATION STUDIES

We conduct an ablation study on the large hybrid setting by comparing the accuracy and formatting reward of our proposed method with the uniform baseline to isolate the effect of our reweighting. The

uniform baseline employs identical hyperparameters, including data sampling and model training pipeline, yet only differs from ours in the loss reweighting mechanism ($\lambda_k = 1/K$). In Figure 6, we present the curves for formatting and accuracy reward during training. In the early phase, the baseline climbs format faster—consistent with format being a low-variance, easy-to-optimize signal—yet a crossover soon appears and our method surpasses it as training progresses (step 40). In contrast, for accuracy, our method opens a growing lead over time (right). The behavior aligns with our scheduler: once the format objective shows fast convergence and low instability, its weight is down-regulated and capacity is reallocated to harder, higher-variance objectives (e.g., accuracy), avoiding over-optimization of formatting while improving task correctness.

We also empirically found that using the same format reward for different domains is suboptimal. We start by examining the approach from Liang et al. (2025) by employing the same thinking reward on every domain, including scene understanding tasks. In Figure 7, we plot the response length on the segmentation task during training and find that the Qwen2.5-VL model rapidly trims its chain-of-thought and answers the question directly, later during training. This behavior suggests that explicit reasoning is unnecessary for such perception tasks and that encouraging long rationales can even be detrimental. We also include qualitative examples in the appendix to show how the model gradually suppresses its reasoning trace during training. Motivated by this observation, in our broader setting, we keep answer-format rewards for perception domains (no thinking) and reserve thinking rewards for tasks that truly benefit from step-by-step reasoning.

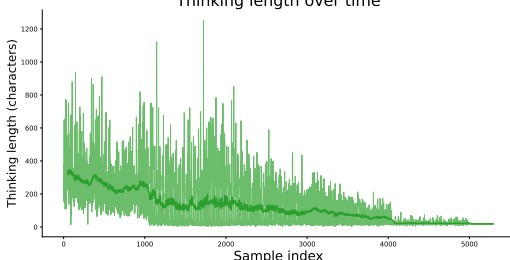 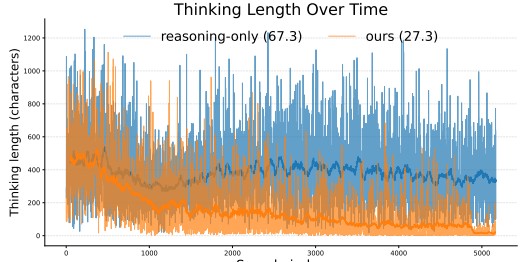

Figure 7: **Thinking length on a segmentation task during finetuning.** When a uniform "thinking reward" is applied to all domains, the model quickly learns that long chains of thought are unnecessary for segmentation. The average response length drops from several hundred characters at the start of training to tens (often near zero) later on.

Figure 8: **Thinking length on a reasoning task during finetuning.** We compare a model trained reasoning-only (blue) against our replay + dynamic reweighting method (orange). Highlighted curves show the running average thinking length per example, where our method generates only $\sim 27.3$ words per question, compared with baseline ($\sim 67.3$).

Figure 8 tracks the length of the generated thinking segment on the reasoning task throughout training. When trained only on reasoning data, the model maintains long chains of thought with high variability. In contrast, mixing general-capability replay with dynamic objective reweighting progressively reduces thinking length and stabilizes variance, converging to concise rationales (60% reduction, $67 \to 27$ words on average) while preserving accuracy. This shorter reasoning directly improves inference efficiency—fewer generated tokens reduce latency and compute cost—without sacrificing problem-solving quality.

## 6 CONCLUSION

In this paper, we investigate the forgetting issues of recent reasoning vision language models and find that those models exhibit clear forgetting of general knowledge obtained during pretraining. Motivated by this, we propose a fix by replaying general data during finetuning and a plug-in method to reweight objectives without the additional cost of training external models. On reasoning benchmarks, our proposed method not only preserves the general knowledge but also improves the target reasoning performance by properly reweighting the rewards of those tasks.

## LIMITATIONS

Our proposed framework is generic and extends naturally beyond RLVR and SFT to preference- and alignment-based objectives (Rafailov et al., 2023; Garg et al., 2025; Hong et al., 2024; Ethayarajh, 2024), process reward models (Lightman et al., 2023a; Setlur et al., 2024). However, due to constraints on the training datasets available for this work, our empirical evaluation focuses on RLVR and standard SFT settings. We expect our method to yield similar gains over uniform or manually tuned baselines with any heterogeneous objective sets, but we leave a comprehensive evaluation across non-RL objectives to future work. In practice, applying our scheduler to non-RL losses does not require an expensive coefficients search or per-objective normalization due to its magnitude-agnostic nature.

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

# A APPENDIX

In this section, we provide detailed statistics of the training datasets used in our experiments, along with implementation details and additional experimental results.

## A.1 DATA STATISTICS AND IMPLEMENTATION DETAILS

Table 3 reports the full statistics of the training corpora used in our experiments. For LLaVA-OneVision-OCR, we extract OCR-focused subsets from the official LLaVA-OneVision release (Li et al., 2024):IIIT5K (Mishra et al., 2012), HME100K (hme, 2021), IAM (Marti & Bunke, 2002), TextCaps (Sidorov et al., 2020), and TextOCR (Singh et al., 2021) alongside release-provided synthetic/curated subsets (rendered_text, k12_printing, chrome_writing). Those images are resized so that the longer side is $\leq 512\,\mathrm{px}$ while preserving aspect ratio to mitigate out-of-memory errors without altering task semantics.

Table 3: **Data statistics of each data source.** We present the original volume of data (# samples).

| Dataset | Domain | Answer Type | Rewards/Objectives | # samples |
|---|---|---|---|---|
| RefCOCO (Kazemzadeh et al., 2014) | Referring Expression Comprehension | 2D Bounding Box | IoU, Answer Format | 321327 |
| LLaVA-OneVision-OCR (Li et al., 2024) | Scene Text-Centric Visual Question Answering | Natural Language | Next Token Prediction | 66468 |
| ThinkLite-VL-70k (Wang et al., 2025b) | Math Reasoning & Natural Image/Chart Understanding | Natural Language | Acc, Thinking Format | 69997 |
| LISA-train (Lai et al., 2023) | Referring Expression | 2D Bounding Box | IoU, Thinking Format | 1326 |
| GeoQAV (Li et al., 2025a) | Math Visual Question Answering | Multiple Choice | Acc, Thinking Format | 1969 |
| SAT-train (Ray et al., 2024b) | Spatial Visual Question Answering | Natural Language | Acc, Thinking Format | 15000 |
| ScienceQA-train (Lu et al., 2022b) | Science Visual Question Answering | Multiple Choice | Acc, Thinking Format | 6218 |

We optimize with GRPO and SFT losses using AdamW (Loshchilov & Hutter, 2017) ($\beta_1$=0.9, $\beta_2$=0.999, $\varepsilon$=$10^{-8}$). The learning rate follows a linear schedule: $10\%$ warmup to $\eta_{\max}$=$1{\times}10^{-6}$, then linear decay to 0. Window size $W$ and temperature $T$ are set to 10 and 5.0, respectively, in our experiments. All runs use bfloat16 precision (Wang & Kanwar, 2019; Micikevicius et al., 2018) and FlashAttention kernels (Dao et al., 2022) for memory- and throughput-efficient attention. We enable thinking mode on reasoning tasks by enforcing structured traces (i.e., wrapping thoughts in `<think>...</think>`). which has been shown to improve reasoning and transparency (Hu et al., 2025b; Xie et al., 2025; Chen et al., 2025). Our evaluation protocol closely follows LMMS-Eval (Zhang et al., 2024a) and VLMEvalKit (Duan et al., 2024).

---

**Evaluation prompt**

**Non-Thinking:**
{Question}
Output the in `<answer> </answer>` tags.
**Thinking:**
{Question}
Output the thinking process in `<think> </think>` and final answer (option) in `<answer> </answer>` tags.

---

We provide the pseudocode for our proposed method in Algorithm 2. To use it, one first computes the task losses, calls `update` to update the task weighting, and then obtains the weighted loss via `get_weighted_loss` to perform standard backpropagation. For typical settings (e.g., Qwen-3B/7B, $K < 10$ objectives, window size $W = 10$), our method introduces only $\Theta(KW)$ extra scalar operations and $\Theta(KW)$ memory, which is negligible compared to the $\Theta\left(10^{11}\right) - \Theta\left(10^{12}\right)$ FLOPs per step of the underlying model; in practice we observed no measurable slowdown.

---

**Algorithm 1** Our proposed method replays general data during RLVR training and adaptively reweights objectives of interest.

---

**Require:** Base parameters $\theta^{(0)}$; domain list $\{\mathcal{D}_n\}_{n=1}^N$ is a union of general $\{\mathcal{D}_1^G, \ldots, \mathcal{D}_M^G\}$ and reasoning domain $\{\mathcal{D}_1^R, \ldots, \mathcal{D}_L^R\}$ ; objectives $\{L_k\}_{k=1}^K$; window size $W$; temperature $T$; total iterations $T_{\max}$

1: Initialize $\lambda_k^{(0)} \leftarrow 1$ for all $k$
2: Initialize loss history buffers $\mathcal{B}_k$ of length $2W$ for each objective $k$
3: **for** $t = 1$ to $T_{\max}$ **do**
4:     Sample mini-batches from reasoning and replay data on each domain $\mathcal{D}_n$
5:     Compute per-domain, per-objective losses $\ell_{n,k}^{(t)}(\theta^{(t)})$
6:     Compute per-objective averaged losses

$$L_k^{(t)} \leftarrow \frac{1}{N} \sum_{n=1}^N \ell_{n,k}^{(t)}(\theta^{(t)}), \quad \forall k$$

7:     **for** $k = 1$ to $K$ **do**
8:         Push $L_k^{(t)}$ into buffer $\mathcal{B}_k$ (FIFO)
9:     **end for**
10:    **if** $t \geq 2W$ **then**
11:      **for** $k = 1$ to $K$ **do**
12:         Compute current-window mean: $\mu_k^{(t)} \leftarrow \frac{1}{W} \sum_{s=t-W+1}^t L_k^{(s)}$ and previous-window mean $\tilde{\mu}_k^{(t)} \leftarrow \frac{1}{W} \sum_{s=t-2W+1}^{t-W} L_k^{(s)}$
13:         Compute the instability:

$$\sigma_k^{(t)} = \sqrt{\frac{1}{2W-1} \sum_{s=t-2W+1}^t \left(L_k^{(s)} - \mu_k^{(t)}\right)^2}$$

14:         Compute the convergence rate $c_k^{(t)} \leftarrow \frac{\tilde{\mu}_k^{(t)}}{\mu_k^{(t)}}$ , the inverse signal-to-noise ratio $i_k^{(t)} \leftarrow \frac{\sigma_k^{(t)}}{\mu_k^{(t)} + \tilde{\mu}_k^{(t)}}$ and the relative priority between domains:

$$s_k^{(t)} \leftarrow c_k^{(t)} + i_k^{(t)}$$

15:      **end for**
16:      Calculate softmax weights:

$$\lambda_k^{(t)} \leftarrow \frac{K \exp\left(s_k^{(t)}/T\right)}{\sum_{j=1}^K \exp\left(s_j^{(t)}/T\right)}, \quad \forall k$$

17:    **else**
18:      $\lambda_k^{(t)} \leftarrow 1$ for all $k$
19:    **end if**
20:    Compute final objective:

$$\mathcal{L}^{(t)}(\theta^{(t)}) \leftarrow \frac{1}{K} \sum_{k=1}^K \lambda_k^{(t)} L_k^{(t)}$$

21:    Update parameters:

$$\theta^{(t+1)} \leftarrow \theta^{(t)} - \eta \nabla_\theta \mathcal{L}^{(t)}(\theta^{(t)})$$

22: **end for**

---

**Algorithm 2** Implementation of our proposed method in PyTorch-like Pseudocode

```
class ObjectivesReweighter:
def __init__(self, num_objectives, window_size, T=5.0):
    # num_objectives  (K,) number of objectives (rewards / losses)
    # window_size  W, length of each averaging window
    self.K = num_objectives
    self.W = window_size
    self.T = T
#     loss_history[k] stores a list of recent scalar losses for
objective k
    self.loss_history = [deque(maxlen=2 * self.W)
                            for _ in range(num_objectives)]
#     current weights λ (no grad, treated as buffer)
    self.lambdas = torch.ones(self.K)
    self.step = 0

def get_weighted_loss(self, losses):
    # losses  (K,) tensor of per-objective losses $L_k^{(t)}$
    # returns $\mathcal{L}^{(t)} = \frac{1}{K} \sum_k \lambda_k^{(t)} L_k^{(t)}$
    weights = self.lambdas.detach()
    loss = (weights * losses).mean()
    return loss

def update(self, losses):
    # losses  (K,) tensor of current per-objective losses (no grad
needed)
    self.step += 1
#     append current losses into history (FIFO of length at most
2W)
    for k in range(self.K):
        self.loss_history[k].append(losses[k].detach())

#     if not enough history, keep uniform mixing
    if self.step < 2 * self.W:
        self.lambdas = torch.ones(self.K)
        return

#     compute per-objective signals $c_k^{(t)}$ and $i_k^{(t)}$
    c = torch.zeros(self.K)
    i = torch.zeros(self.K)
    for k in range(self.K):
        hist = torch.stack(self.loss_history[k])
        recent = hist[-self.W:] # current window
        old = hist[-2*self.W:-self.W] # previous window
        mu = recent.mean()
        mu_old = old.mean()
        sigma = hist.std(unbiased=True)
#         $c_k^{(t)} = \mu_k^{(t)} / \tilde{\mu}_k^{(t)}$
        c[k]  = mu_old / mu
#         $i_k^{(t)} = \sigma_k^{(t)} / (\mu_k^{(t)} + \tilde{\mu}_k^{(t)})$
        i[k]  = sigma / (mu + mu_old)

#     priority scores $s_k^{(t)} = c_k^{(t)} + i_k^{(t)}$
    s = c + i
#     temperature-controlled softmax, normalized so $\frac{1}{K} \sum_k \lambda_k = 1$
    w = torch.softmax(s / self.T, dim=-1)
    self.lambdas = self.K * w.detach()
```

## A.2 PROMPTS USED IN OUR EXPERIMENTS

The prompt used for training our model is shown in Figure 9.

**Example of question**

**Cups of lemonade sold**

| Day | Number of cups |
|---|---|
| Saturday | 82 |
| Sunday | 96 |
| Monday | 82 |
| Tuesday | 93 |
| Wednesday | 88 |

**Image**

**Question:** Julie wrote down how many cups of lemonade she sold in the past 5 days. What is the range of the numbers?

**Prompt:** Julie wrote down how many cups of lemonade she sold in the past 5 days. What is the range of the numbers? Output the thinking process in <think> </think> and final answer in <answer> </answer> tags.

**Prompt**

<think>
To find the range of the numbers, we need to follow these steps:
1. Identify the highest number in the list.
2. Identify the lowest number in the list.
3. Subtract the lowest number from the highest number.

The numbers from the table are: 82, 96, 82, 93, 88.

Step 1: Identify the highest number.
The highest number is 96.

Step 2: Identify the lowest number.
The lowest number is 82.

Step 3: Subtract the lowest number from the highest number.
96 - 82 = 14

Therefore, the range of the numbers is 14.
</think>
**<answer>14</answer>**

**Answer**

**Ground truth: 14**

Figure 9: **Example conversation for finetuning** We ask the model to produce its reasoning chain within <think> ... </think> and the final answer in <answer>...</answer> tags.

### A.3 QUALITATIVE EXAMPLES

Responses from models trained with reasoning data only 10 and when including general data.

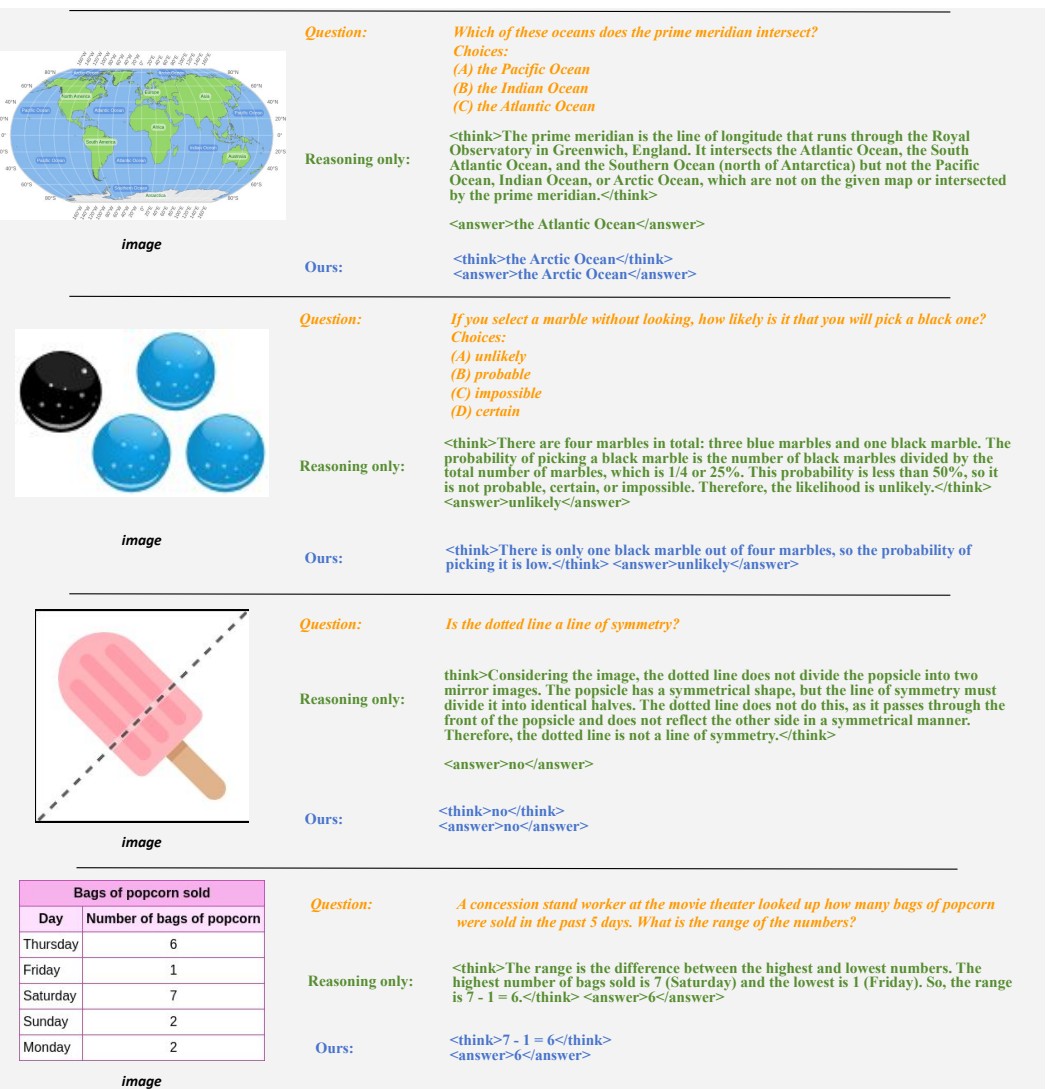

Figure 10: **Generated answers from the reasoning-only finetuned model and our model.** While the reasoning-only finetuned model generates long reasoning chains for simple visual questions, our model produces more concise answers, especially on non-math tasks.

### A.4 EVOLUTION OF ROLLOUTS DURING TRAINING WITH THE THINKING REWARD

Below, we show generated responses from the model at different points during training, where we can observe the finetuned model gradually shortening its chain of thought on image-understanding tasks to provide final answers directly. This motivates our use of non-thinking rewards on tasks that do not require reasoning, instead of employing the same thinking-format reward for every task (Liang et al., 2025).

**Example 1:** rollout # 0

**Prompt:** Please provide the bounding box coordinate of the region this sentence describes: flicking tail.
Output the thinking process in `<think>` `</think>` and final answer in `<answer>` `</answer>` tags. The output answer format should be as follows:
`<answer>[x1, y1, x2, y2]</answer>`
**Answer:** To determine the bounding box coordinates for the zebra that is flicking its tail, we need to follow these steps:
1. Identify the zebra in the image that is flicking its tail.
2. Observe the position and direction of the zebra's tail.
3. Define a bounding box that encapsulates the entire zebra, keeping in mind that the tail should be included in the box.
Upon inspection:
- The zebra on the right is flicking its tail, as evident from the blurred tail motion and the position of the tail.
- The bounding box should encompass the entire zebra, including its tail.
`<answer>[291, 41, 432, 315]</answer>`

**Example 2:** rollout # 500

**Prompt:** Please provide the bounding box coordinate of the region this sentence describes: second from right.
Output the thinking process in `<think>` `</think>` and final answer in `<answer>` `</answer>` tags. The output answer format should be as follows:
`<answer>[x1, y1, x2, y2]</answer>`
**Answer:** `<think>`To determine the bounding box for the second from right, I first identify the rightmost object, which is the one on the far right, identified by its position and bounding box. The second from right is the one to the left of the rightmost object. By visual inspection, the bounding box for the second from right should start after the farthest object and extend leftward.`</think>`
`<answer>[237, 72, 374, 328]</answer>`

**Example 3:** rollout # 1000

**Prompt:** Please provide the bounding box coordinate of the region this sentence describes: boat blocked by cow stomach.
Output the thinking process in `<think>` `</think>` and final answer in `<answer>` `</answer>` tags. The output answer format should be as follows:
`<answer>[x1, y1, x2, y2]</answer>`
**Answer:** `<think>` The sentence refers to the boat that is partially covered by the cow's stomach. This boat seems to be the one with the green and white colors.`</think>`
`<answer>[327,21,631,280]</answer>`

**Example 4:** rollout # 5000

**Prompt:** Please provide the bounding box coordinate of the region this sentence describes: guy on left with backpack on.
Output the thinking process in `<think>` `</think>` and final answer in `<answer>` `</answer>` tags. The output answer format should be as follows:
`<answer>[x1, y1, x2, y2]</answer>`
**Answer:** `<think>[114,53,236,378]</think>`
`<answer>[114,53,236,378]</answer>`

## A.5 REWARD VALUES AT THE END OF TRAINING

Similar to what we observe in the main paper, Figure 11 showcases the final performance of our model and the uniform baseline. Results show near-parity on thinking formatting (`<think></think>` `<answer> </answer>`) and direct answer reward `<answer> </answer>`) but consistent improvements on reasoning score, IoU and mean token accuracy (+2.01, +1.11, and +1.40 points respectively). This aligns with our design goal: once format signals saturate, down-weight them and shift capacity to harder, higher-variance objectives, improving accuracy while maintaining output format.

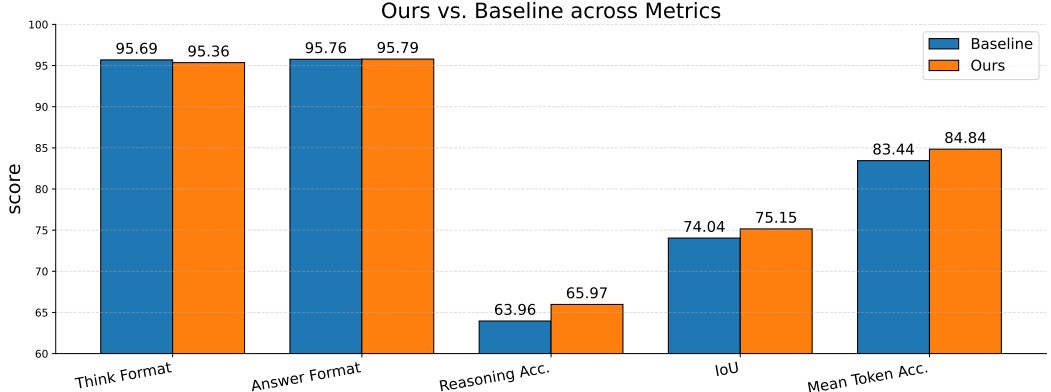

Figure 11: **Final performance across metrics.** We compare a uniform baseline with our dynamic reweighting. The gains on correctness-oriented metrics indicate that reallocating weight away from saturated format rewards toward harder objectives yields better solutions without sacrificing adherence to templates.

In Figure 12, we plot the coefficients of the five objectives used in the hybrid setup. From these, we can rank them by how strongly our method focuses on each objective, from low to high: format rewards, IoU reward, next-token prediction (on the OCR task), and reasoning accuracy.

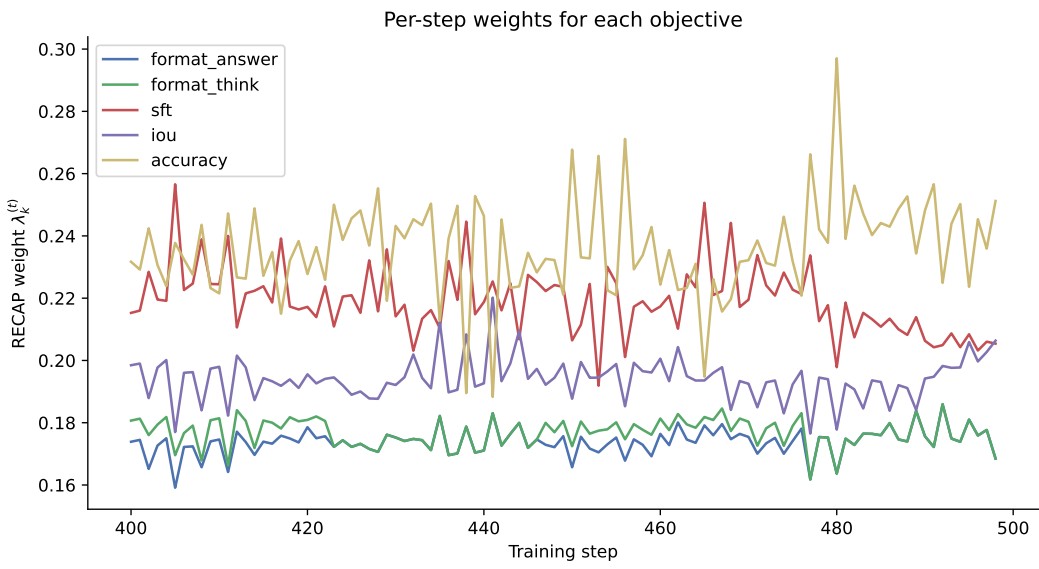

Figure 12: **Evolution of per-objective coefficients.** In the last 100 iterations, the coefficient for each objective is relatively consistent, with format rewards receiving the lowest focus while the supervised finetuning objective and accuracy rewards are emphasized due to their instability.

## A.6 Reward dynamics during training

Given the multi-objective nature of the problem, one might apply existing methods in multi-task learning literature (Guo et al., 2025) for reweighting different objectives and rewards. In practice, this is difficult for two reasons. First, computing per-objective gradients is prohibitively expensive at LLM scale, especially under reinforcement learning. Second, on-policy RL signals are high-variance and non-stationary (Henderson et al., 2018), making per-iteration statistics unreliable indicators of task progress. As shown in Figure 13, all rewards fluctuate substantially within their $[0, 1]$ range, with the standard deviation of the total reward peaking near 0.9 around step $\sim 20$. Thus, we propose a method that utilizes a sliding window, which provides a more robust proxy for understanding convergence behavior.

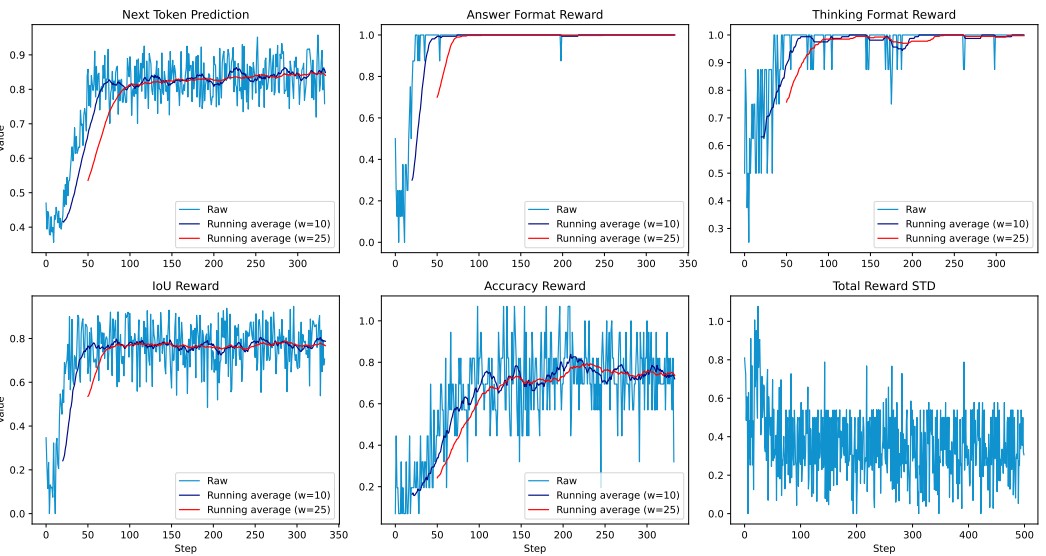

Figure 13: **Reward dynamics and variability during RLVR training.** Per-step rewards (light traces) and sliding-window means (dark curves) for six metrics: Next-Token Prediction, Answer-Format, Thinking-Format, IoU, Accuracy, and the Total-Reward Standard Deviation (lower-right). Asynchronous convergence and high variance motivate short-horizon statistics for dynamic objective reweighting rather than per-iteration magnitudes.

We also conduct an ablation on the effect of the window size $W$ by increasing it from 10 (our default throughout the experiments) to larger values, up to 25, as shown in Figure 13. Since we train Qwen2.5-VL-7B for 500 iterations, setting $W = 25$ delays the onset of dynamic reweighting by 50 iterations, according to Algorithms 1 and 2, because our method requires $2W$ steps of history. By the time reweighting becomes active, some rewards have already entered a near-converged regime, which reduces the usefulness of the convergence-rate term and makes the scheme rely mostly on the instability term. We therefore choose $W = 10$ as a reasonable compromise between sensitivity and robustness: it accumulates enough information while remaining responsive to the current state of training.

To better understand the effectiveness of our method, we use a slightly modified version of the two-task toy example provided in (Liu et al., 2021b). The two tasks $L_1(x)$ and $L_2(x)$ are defined on $x = (x_1, x_2)^\top \in \mathbb{R}^2$,

$$L_1(x) = f_1(x)g_1(x) + f_2(x)h_1(x) + 3\epsilon$$
$$L_2(x) = f_1(x)g_2(x) + f_2(x)h_2(x),$$

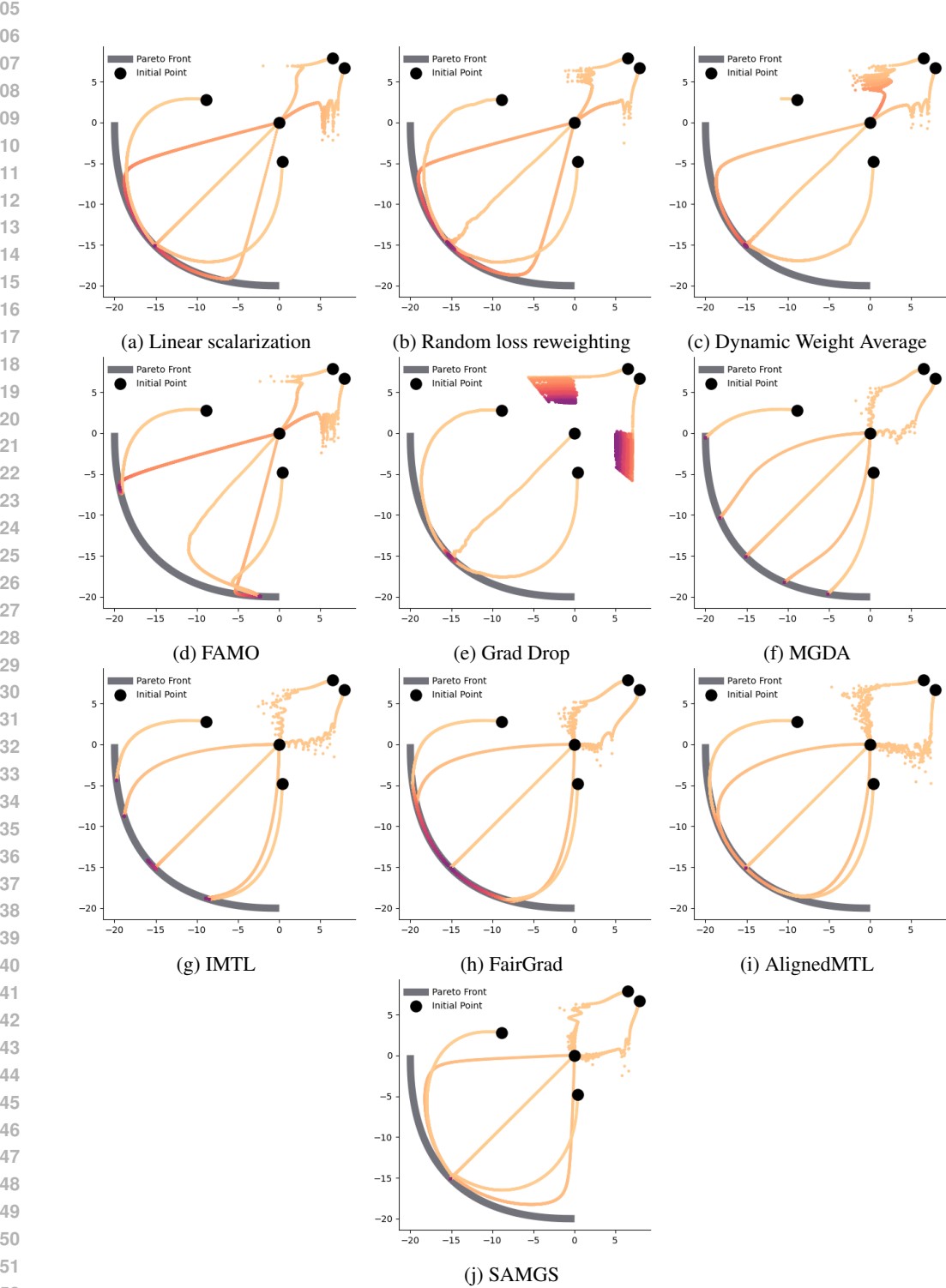

(a) Linear scalarization     (b) Random loss reweighting     (c) Dynamic Weight Average

(d) FAMO     (e) Grad Drop     (f) MGDA

(g) IMTL     (h) FairGrad     (i) AlignedMTL

(j) SAMGS

Figure 14: **Additional illustrative experimental results** Introducing noise in the first objective causes instability among many MTL methods. Methods that leverage per-step loss magnitude statistics like FAMO, DWA and UW exhibit considerable unstable convergence on different initializations.

where $\epsilon \sim N(0,1)$ the functions are given by:

$$f_1(x) = \max\big(\tanh(0.5x_2), 0\big)$$

$$f_2(x) = \max\big(\tanh(-0.5x_2), 0\big)$$

$$g_1(x) = \log\Big(\max\big(|0.5(-x_1 - 7) - \tanh(-x_2)|, 0.000005\big)\Big) + 6$$

$$g_2(x) = \log\Big(\max\big(|0.5(-x_1 + 3) - \tanh(-x_2) + 2|, 0.000005\big)\Big) + 6$$

$$h_1(x) = \big((-x_1 + 7)^2 + 0.1(-x_1 - 8)^2\big)/10 - 20$$

$$h_2(x) = \big((-x_1 - 7)^2 + 0.1(-x_1 - 8)^2\big)/10 - 20.$$

We use five different starting points $\{(-8.5, 7.5), (0, 0), (9.0, 9.0), (-7.5, -0.5), (9.0, -1.0)\}$. Those points are optimized by Adam (Kingma, 2015) with a learning rate of 1e-2 for 10000 iterations. We include loss balancing methods such as UW (Cipolla et al., 2018), DWA (Liu et al., 2019), GradNorm (Chen et al., 2018), and RGW (Lin et al., 2022), FAMO (Liu et al., 2023) and gradient-based methods: PCGrad (Yu et al., 2020), CAGrad (Liu et al., 2021a), GradDrop (Chen et al., 2020), MGDA (Dong et al., 2015), IMTL (Liu et al., 2021c), Nash-MTL (Navon et al., 2022), Aligned-MTL (Senushkin et al., 2023) and SAM-GS (Borsani et al., 2025). Their convergence behaviors are presented in Figure 5 in the main paper and Figure 14 in this appendix, from which we can see improvements across all initialized solutions over other gradient-based methods while being much more efficient.

Table 4: **Evaluation results on NYUv2 scene understanding.** Test performance for three tasks: semantic segmentation, depth estimation, and surface normal. We highlight the best loss-magnitude based MTL method in **bold** and gradient-based MTL method by underscore.

| Complexity | | Segmentation | | Depth | | Surface Normal | | | | | | $\Delta$m% $\downarrow$ |
|---|---|---|---|---|---|---|---|---|---|---|---|---|
| | | | | | | Angle Distance $\downarrow$ | | Within $t°$ $\uparrow$ | | | | |
| | | mIoU $\uparrow$ | Pix Acc $\uparrow$ | Abs Err $\downarrow$ | Rel Err $\downarrow$ | Mean | Median | 11.25 | 22.5 | 30 | | |
| | STL | 38.30 | 63.76 | 0.6754 | 0.2780 | 25.01 | 19.21 | 30.14 | 57.20 | 69.15 | | |
| | LS | 39.29 | 65.33 | 0.5493 | 0.2263 | 28.15 | 23.96 | 22.09 | 47.50 | 61.08 | | 5.59 |
| | SI | 38.45 | 64.27 | 0.5354 | 0.2201 | 27.60 | 23.37 | 22.53 | 48.57 | 62.32 | | 4.39 |
| | RLW | 37.17 | 63.77 | 0.5759 | 0.2410 | 28.27 | 24.18 | 22.26 | 47.05 | 60.62 | | 7.78 |
| $\Theta(1)$ | DWA | 39.11 | 65.31 | 0.5510 | 0.2285 | 27.61 | 23.18 | 24.17 | 50.18 | 62.39 | | 3.57 |
| | UW | 36.87 | 63.17 | 0.5446 | 0.2260 | **27.04** | 22.61 | 23.54 | 49.05 | 63.65 | | 4.05 |
| | Ours | **41.26** | **66.79** | **0.5303** | **0.2203** | 27.11 | **22.23** | **24.64** | **50.88** | **64.02** | | **0.77** |
| | GradNorm | 20.09 | 64.64 | 0.7200 | 0.2800 | 24.83 | 18.86 | 30.8 | 57.94 | 69.73 | | 7.22 |
| | MGDA | 30.47 | 59.90 | 0.6070 | 0.2555 | 24.88 | 19.45 | 29.18 | 56.88 | 69.36 | | 1.38 |
| | PCGrad | 38.06 | 64.64 | 0.5550 | 0.2325 | 27.41 | 22.80 | 23.86 | 49.83 | 63.14 | | 3.97 |
| $\Theta(K)$ | GradDrop | 39.39 | 65.12 | 0.5455 | 0.2279 | 27.48 | 22.96 | 23.38 | 49.44 | 62.87 | | 3.58 |
| | CAGrad | 39.79 | 65.49 | 0.5486 | 0.2250 | 26.31 | 21.58 | 25.61 | 52.36 | 65.58 | | 0.20 |
| | IMTL-G | 39.35 | 65.60 | 0.5426 | 0.2256 | 26.02 | 21.19 | 26.2 | 53.13 | 66.24 | | −0.76 |
| | Nash-MTL | 40.13 | 65.93 | 0.5261 | 0.2171 | 25.26 | 20.08 | 28.4 | 55.47 | 68.15 | | −4.04 |

Table 4 reports the performance of different MTL methods on the real-scene understanding benchmark, which includes one segmentation task and two pixel-level regression tasks. Overall, our method nearly matches the single-task baselines ($\Delta$m% $\downarrow \approx 0$) while being $K = 3$ times more efficient in both runtime and memory, and it consistently outperforms all other loss-reweighting methods across all metrics (except Angle Distance Mean, where it is competitive with Uncertainty Weighting). Notably, our approach even surpasses several established gradient-based methods, such as GradNorm, MGDA, PCGRAD, and GradDrop, while remaining three times faster. We also observe a clear Pareto trade-off: although NashMTL achieves the highest overall relative improvement in $\Delta$m% $\downarrow$, it lags behind GradNorm and MGDA on the surface-normal task, whereas these methods incur substantial performance drops on segmentation and depth estimation.

Figure 15 plots the loss curves for three different objectives, showing stable optimization across all of them. In contrast, our RL rewards are much sparser than in this SFT setting, and the training curves in Figures 4 and 13 exhibit substantially higher fluctuations. This motivates a more robust loss-reweighting mechanism, as relying solely on instantaneous per-step loss values is not sufficiently representative of the underlying learning dynamics or objective progress.

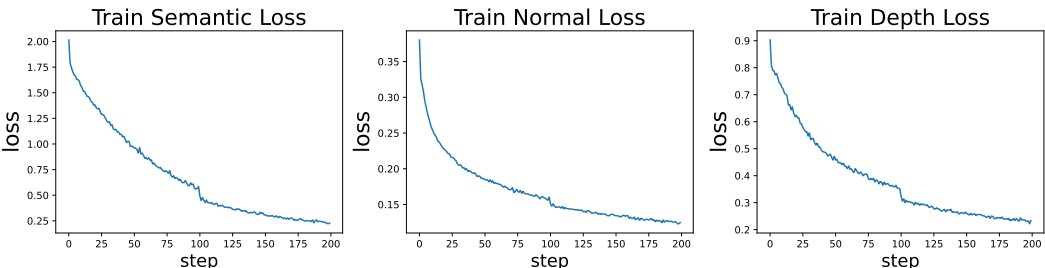

Figure 15: **Loss curves during training on NYUv2.** Compared to the training curves in our experiments (e.g. Figure 13), those curves in this experiment are much more smooth and stable, where per-step statistics can provide informative signal of the learning progress.

The running-time comparison in Figure 16 shows that, although effective in some scenarios, gradient-based MTL methods require storing and computing all task gradients, incurring $\Theta(K)$ space and time overhead where (K) is the number of objectives. In our illustrative setup with (K=3), this already makes these methods about three times slower ($\sim 300s$ vs. $\sim 100s$) than single-task baselines and other loss-reweighting approaches. In our main RLVR experiments, we have 4 domains with 2 objectives per domain (K=8), which would make gradient-manipulation methods roughly $8\times$ slower than standard training. For this reason, we focus on loss-reweighting mechanisms, which avoid such substantial computational overhead.

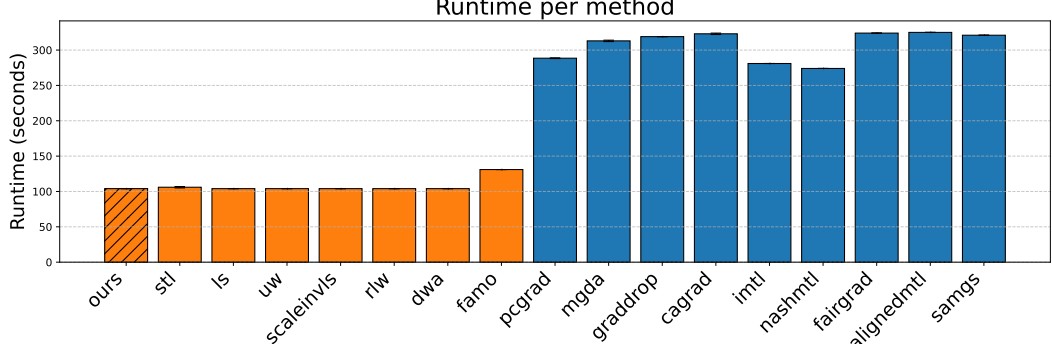

Figure 16: **Running time of different MTL methods.** While being robust to noise in some scenarios, gradient-based methods (denoted by blue) often cause significant overhead ($\approx k$ times as they compute per-objective gradients) compared to loss-magnitude based methods (denoted by orange).

We conduct ablation studies on the temperature hyperparameter $T$ and the trade off $\alpha$ between the convergence rate and the inverse signal-to-noise ratio: $s_k^{(t)} = \alpha c_k^{(t)} + (1 - \alpha) i_k^{(t)}$. From Figure 17, we observe that intermediate values such as $\alpha = 0.5$ or $0.75$ strike a good balance between the two terms and yield noticeably more stable convergence across all initializations.

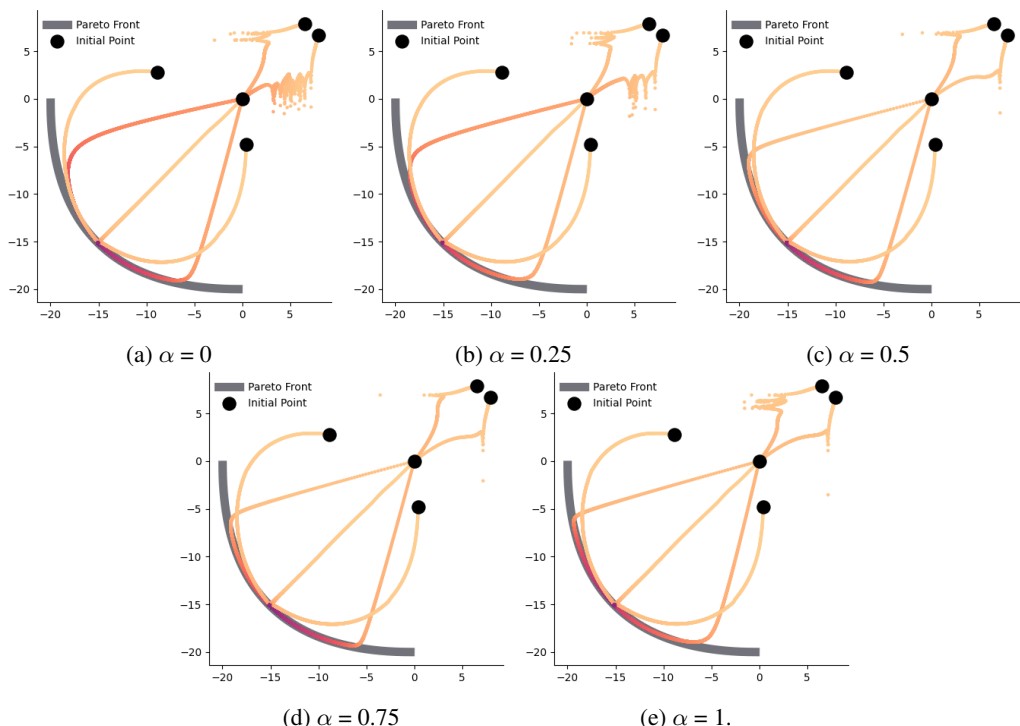

(a) $\alpha = 0$      (b) $\alpha = 0.25$      (c) $\alpha = 0.5$

(d) $\alpha = 0.75$      (e) $\alpha = 1$.

Figure 17: **Ablation on the trade-off** $\alpha$. Using only the convergence rate ($\alpha = 1$) or only the inverse signal-to-noise ratio ($\alpha = 0$) leads to unstable learning on the second and first objective, respectively.

For the temperature, setting $T$ too low makes training unstable: as shown in Figure 18a, the trajectories exhibit strong fluctuations near the Pareto front. Conversely, setting $T$ to a high value (e.g., $T = 30$) also harms convergence: for the two initializations farthest from the Pareto front, optimization requires many more steps to approach the front (the trajectories remain red for longer).

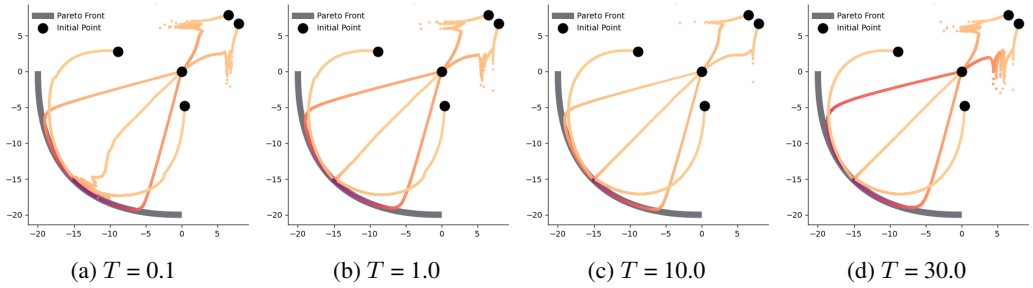

(a) $T = 0.1$      (b) $T = 1.0$      (c) $T = 10.0$      (d) $T = 30.0$

Figure 18: **Ablation studies on the temperature** $T$. $T > 1$ acts as a regularization to avoid extreme reweighting (one domain dominates others) and stabilizes the training.

Due to the large computational requirements of RL training, we find that setting $T = 5$ and $\alpha = 0.5$ works reasonably well in the RLVR-only setting. For simplicity, we keep this configuration for the Hybrid setup and do not perform additional hyperparameter tuning in the large-scale setting. Table 5 reports the results when varying the trade-off $\alpha$, the temperature T, and the window size W. Although upweighting the instability term can increase the weight assigned to the accuracy reward since this term is highly fluctuating, it comes at the cost of sacrificing essential perception skills. For example, $\alpha = 0.25$ improves performance on SAT and ScienceQA by 0.1% and 0.6%, respectively, but reduces ChartQA and InfoVQA performance by 3%. Similar to our illustrative example, decreasing the temperature induces higher variation across tasks—for instance, it yields the highest score on MathVista while reducing SAT performance by 3.2%.

Table 5: **Benchmark results in RLVR-only setting.** Ablation results when varying the temperature and convergence rate-instability trade-off.

| Model | SAT | ScienceQA | MathVista (mini) | ChartQA | InfoVQA | MMMU |
|---|---|---|---|---|---|---|
| MoDoMoDo | 50.0 | 65.7 | 32.2 | 70.4 | 59.9 | 39.1 |
| Ours | 55.2 | 71.6 | 33.2 | 70.4 | 60.8 | 42.4 |
| $\alpha = 0.50$, T = 1.0, W=10 | 52.0 | 71.6 | 33.4 | 68.1 | 58.5 | 40.4 |
| $\alpha = 0.75$, T = 5.0, W=10 | 54.4 | 71.2 | 32.9 | 70.0 | 60.7 | 41.0 |
| $\alpha = 0.25$, T = 5.0, W=10 | 55.3 | 72.2 | 33.7 | 66.1 | 56.8 | 39.3 |
| $\alpha = 0.50$, T = 5.0, W=50 | 51.9 | 70.5 | 32.9 | 69.9 | 59.7 | 40.8 |