# OpenReview forum: "Beyond Reasoning Gains: Mitigating General Capabilities Forgetting in Large Reasoning Model"
_ICLR.cc/2026/Conference — ICLR 2026 Conference Withdrawn Submission_

### Official Review · Reviewer_NAE8 · 2025-10-21

**Soundness:** 1
**Presentation:** 1
**Contribution:** 1
**Rating:** 0
**Confidence:** 3

**Summary:**

I wasn’t able to clearly understand the main concept of the paper (see page 4). Please refer to the weaknesses mentioned below.

**Strengths:**

I cannot mention any strengths at the moment.

**Weaknesses:**

This paper does not yet appear ready for formal review.

The main idea is not clearly articulated, The radar plot (Fig. 1) fails to effectively convey the authors’ intended message—alternative visualization methods could be more appropriate. The overview plot is also difficult to interpret. In addition, there are several stylistic and formatting issues: sentences often lack proper punctuation (e.g., missing periods), and citation formatting should follow standard conventions (e.g., using \citep and \citet appropriately).
Figure 7 in the appendix appears poorly constructed and lacks coherence. The question, prompt, and reasoning process are inconsistent, which raises concerns about the validity of the example.

**Questions:**

How are the convergence rate and instability incorporated into your model formulation, and how do your overall training objectives differ from those in previous RLVR work?

**Details Of Ethics Concerns:**

Figure 7 in the appendix appears poorly constructed and lacks coherence. The question, prompt, and reasoning process are inconsistent, which raises concerns about the validity of the example.

---

> ### Author Response · Authors · 2025-11-26
> **Response to Reviewer NAE8**
>
> We thank the reviewer for your feedback. We have revised the manuscript to clarify the main idea, improve the visualizations, and address the concerns about Fig. 7.
>
> - **Clarifying the main concept and formulation**: We have added an explicit algorithmic description of our method in the main text and a step-by-step Python-style pseudocode in the appendix. These additions are intended to make the core idea of our method easier to follow, in addition to the method section on page 4.
>
> - **The radar plot** As mentioned in our general response, the apparent collapse in performance for some models and their base models is intentionally shown to highlight that our method is effective at preserving general capabilities during reasoning finetuning. We provide detailed numbers in the general response to support the same takeaway in the main paper: while MiMo-VL-7B-RL can preserve the original base model performance quite consistently across different benchmarks, other model families often suffer from catastrophic forgetting of these core skills.
>
> - **Overview figure**: Thank you for the suggestion. We have further updated the overview figure to more clearly illustrate how we (i) replay general-capability data, (ii) compute their corresponding objectives, and (iii) combine them with the reasoning objectives via dynamically updated coefficients to form the final loss. We hope this makes the method easier for readers to follow, and we would be happy to further refine the figure if you have additional suggestions.
>
> - **Typo and the example figure**: Thank you for pointing out the stylistic issues, we have fixed them added image to the Figure 7 (it is now Figure 9) and also additionally provided four more example conversations in a new Figure 10, to give a more representative and coherent picture of the behaviors we discuss in the main text.
>
> - **“Flag for Ethics Review” note**: To the best of our understanding, our work does not raise specific ethical risks beyond those typically associated with training and evaluating large language models. Our examples in Figs. 9–10 do not contain personal or sensitive information and are either synthetic or derived from standard benchmarks. We will be happy to further clarify any specific ethical concern you may have.

---

### Official Review · Reviewer_AKYp · 2025-10-26

**Soundness:** 2
**Presentation:** 2
**Contribution:** 1
**Rating:** 2
**Confidence:** 4

**Summary:**

The paper focuses on the point that RLVR enhances reasoning but often leads to degradation in general capabilities. To address this, authors propose integrating general capability data into RLVR pipeline and dynamically reweighting objective based on window-based estimation of convergence rate and SNR. Authors argue that this allows shifting focus to underperforming objectives. The experimental results show that this strategy leads to an improvement.

**Strengths:**

- The paper probed to empirically show that RLVR actually often degrades general capabilities, which is an interesting point.
- Shows improved performance over a number of benchmarks.

**Weaknesses:**

- The method design feels somewhat rudimentary; for example,
  - Why use window based method instead of exponential moving statistics? Window based method may suffer from abrupt changes when some important statistics leave the window.
  - What motivates prioritizing the objectives that converges fast? If we prioritize the objectives that converges fast, it would learn one objective at a time, as illustrated in the paper; in Figure 3, it is said that the format is optimized first, and the other objectives are learned later. While the paper mentions catastrophic forgetting and continual learning, I believe slowing down the objectives that converges fast so that every objectives have similar rate of convergence is the best way to reduce catastrophic forgetting.
  - While it worked to simply add $c$ and $i$ for the proposed dataset/model/hyperparameters, it is probable that there will be a setting where simple addition would not work. For example, $c$ is correlated with window size but $i$ is not (increasing $W$ will result in decreased $c$). batch size, learning rate, and all these optimization hyperparameters depend on one of two, so if we tweak the hyperparameters, then it is likely the scale of two will be greatly varied.

- There is not enough validation on the proposed reweighting scheme. Over the number of baselines in experiments, the baseline that can explain the impact of proposed reweighting objective is only (default vs ours). While this shows significant performance difference, the design choices are not fully empirically validated; for example, what is the more important factor, SNR or convergence rate? how does varying window size affect the performance? how does temperature affect the performance?

- Other than the proposed rudimentary reweighting scheme that is not sufficiently validated empirically, the contribution of the paper is about including and incorporating general capability dataset and losses. However, I believe that this is not very novel, as such multi-task joint training has been often done in many fields of machine learning, going back to early 2000s.

- some minor comments:
  - estimated old loss averages from $t-W+1$ to $t$ and estimated current loss averages from $t-2W+1$ to $t-W$? is this flipped?
  - How do we sum from $t-W+1$ to $t+W$, where the current timestep is $t$? (when computing $\sigma_k^{(t)}$)

**Questions:**

- Compared to "Default", each objective in "ours" can only have at most $K$ times larger portions in the overall losses. Does the proposed reweighting objective still outperforms the non-reweighted version if we train non-reweighted version $K$ time longer?

---

> ### Author Response · Authors · 2025-11-26
> **Response to Reviewer AKYp**
>
> Thank you very much for your thorough and thoughtful comments and suggestions. We present additional experiments and clarification regarding the reviewer’s concerns and questions below.
>
> **W1**:
> - **The use of sliding window**: We intentionally use non-overlapping sliding windows to estimate the “current” and “previous” loss levels. This gives the convergence rate a clean interpretation as the ratio between the mean loss over the last 𝑊 updates and the preceding 𝑊  updates, which is harder to obtain with exponential moving averages that blend all history into a single statistic. We view the finite window as a feature rather than a limitation as it also provides a well-defined horizon  and avoids long-term inertia from early, highly non-stationary phases of RL training so very early losses quickly become outdated and no longer representative of the current optimization stage. Eventhough, the window size 𝑊 directly controls how far back we look—larger windows smooth more aggressively and integrate longer horizons, while smaller windows react faster to recent changes. This provides an interpretable, tunable trade-off between stability and responsiveness that is well aligned with the non-stationary nature of RLVR training. While window-based methods do drop older samples once they leave the window, the induced changes in the means are scaled by 1/𝑊 and further smoothed by the temperature-controlled softmax. Conceptually, our method is agnostic to the specific estimator, and an EMA-based variant could a straightforward alternative while it might require tuning the EMA coefficient and futher formula to compute the convergence rate.
> - **Prioritize the objectives that converges fast**: We would like to clarify that our scheme downweights objectives that have already converged and prioritizes those that converge more slowly. This prevents easy-to-optimize objectives (e.g., format rewards) from dominating the training signal. You are right, we are solving catastrophic forgetting by replaying general data and balancing the learning across objectives.
> -  **simply add c and i**: In our initial RLVR-only experiments, we indeed considered a weighted combination $s = \alpha c + (1-\alpha)i$ and even designed a scheduler for $\alpha$ to reduce hyperparameter tuning effort. Our goal was to avoid introducing yet another sensitive hyperparameter for RL training, especially at larger scales (e.g., 7B models). Empirically, we found that a simple configuration with $\alpha = 0.5$, temperature $T = 5.0$, and window size $W = 10$ works robustly in the RLVR setting and transfers reasonably well to the Hybrid setup. For this reason, we opted to fix $s = c + i$ and reuse this configuration when training the 7B model. Below we provide the ablation studies for the trade off hyperparameters.
>
> | Model                      | SAT  | ScienceQA | MathVista (mini) | ChartQA | InfoVQA | MMMU |
> |---------------------------|------|-----------|------------------|---------|---------|------|
> | MoDoMoDo                  | 50.0 | 65.7      | 32.2             | 70.4    | 59.9    | 39.1 |
> | Ours (α = 0.50, T = 5.0, W=10)                  | 55.2 | 71.6 | 33.2 | 70.4 | 60.8 | 42.4 |
> | α = 0.50, T = 1.0, W=10   | 52.0 | 71.6      | 33.4             | 68.1    | 58.5    | 40.4 |
> | α = 0.75, T = 5.0, W=10   | 54.4 | 71.2      | 32.9             | 70.0    | 60.7    | 41.0 |
> | α = 0.25, T = 5.0, W=10   | 55.3 | 72.2      | 33.7             | 66.1    | 56.8    | 39.3 |
> | α = 0.50, T = 5.0, W=50   | 51.9 | 70.5      | 32.9             | 69.9    | 59.7    | 40.8 |
>
>
>
> **W2, W3**: Your concerns about empirical validation are very insightful. In response, we have added a dedicated Section A.6 in the appendix that provides a comprehensive comparison against both established and recent multi-task learning and loss-reweighting methods. We believe our approach is novel in this space. Specifically, we find that our method can even outperform other MTL approaches when noise is introduced in our illustrative experiment, highlighting the benefit of our sliding window design and the noise-aware inverse SNR term. On established MTL benchmarks, our method achieves the best performance among loss-reweighting baselines and is competitive with, and often surpasses, more complex gradient-based MTL methods.
>
>
> **C**: Thank you for pointing this out. You are correct that the “old” and “new” losses were inadvertently flipped due to a typo, and that the standard deviation should be computed over the entire window from $t-2W+1$ to $t$
>
> **Q**: In this particular experiment, we are training a 7B model with reinforcement learning using K=5 objectives. Thus, training for 𝐾 time longer would incur a significant computational overhead (for example, even the smaller setup already requires about one day on 8×A100 80GB GPUs). For all experiments in the main text and the appendix, we keep the model size and the number of training iterations fixed across methods to ensure a fair comparison.

---

### Official Review · Reviewer_QzJe · 2025-11-01

**Soundness:** 4
**Presentation:** 3
**Contribution:** 3
**Rating:** 6
**Confidence:** 4

**Summary:**

This paper investigates catastrophic forgetting of general capabilities in vision-language models (VLMs) when fine-tuned using Reinforcement Learning with Verifiable Rewards (RLVR) for reasoning tasks.

The authors demonstrate that RLVR improves reasoning but degrades perception and robustness. They propose a replay-based strategy with dynamic objective reweighting that monitors convergence rates and instability of individual objectives, automatically down-weighting saturated objectives (e.g., format compliance) while focusing on harder ones (e.g., reasoning accuracy).

Experiments on Qwen2.5-VL-3B/7B models show the method preserves general capabilities while improving reasoning performance.

**Strengths:**

Originality: The dynamic reweighting based on convergence rate and instability signals is novel. The observation that different rewards (format, accuracy, IoU) converge at different rates with varying stability is insightful and well-motivated by Figure 3.

Quality: Solid experimental validation across two model scales with comprehensive benchmarking on reasoning (MMMU-PRO, MathVista), perception (LISA, OCRBench), and robustness tasks. Good ablation studies isolating the reweighting contribution, and practical comparisons showing 60% reduction in thinking length without sacrificing accuracy.

Clarity: Well-written with clear motivation. Figures 1-3 effectively illustrate the problem and solution. The mathematical formulation is straightforward.

Significance: Addresses a critical practical problem for deploying reasoning models. The solution requires no additional model training (unlike MoDoMoDo) and integrates easily into existing RLVR pipelines.

**Weaknesses:**

1. Weak theoretical justification: The choice s_k = c_k + i_k appears ad-hoc without justification. Why not weighted combinations, gradient magnitudes, or loss curvature? The unweighted sum needs empirical or theoretical support.

2. Incomplete experimental details:
* Window size W not specified
* Exact composition and proportion of "general data" unclear
* No discussion of reward normalization across different scales
* No error bars or multiple runs despite RL stochasticity

3. Limited failure analysis: On VizWiz, the method (61.97) underperforms Coreset (63.76) and Reasoning-only (62.45). No analysis of when/why dynamic reweighting fails. Understanding these cases would strengthen the work.

4. Computational overhead not reported: No wall-clock time or memory overhead analysis. Computing running statistics and reweighting adds complexity—how significant is this in practice?


### Minor Issues
* Figure 1 radar plots could be higher quality
* Some notation inconsistencies (D for domains vs. datasets)
* Related work could better connect to multi-objective RL and multi-task learning literature

**Questions:**

1. Can you provide sensitivity analysis for window size W and temperature T?
2. How are rewards normalized across different scales (IoU vs. accuracy vs. format)?
3. Why does the method underperform on VizWiz? Can you characterize when dynamic reweighting helps vs. hurts?
4. Can you compare against well-tuned KL regularization more thoroughly? LwF achieves competitive results in Table 1.
5. Have you tried alternative formulations beyond s_k = c_k + i_k, like s_k = α·c_k + β·i_k? Can you provide justification for this specific choice?
6. Do the weights eventually converge to stable values, or keep fluctuating throughout training?

---

> ### Author Response · Authors · 2025-11-25
> **Response to Reviewer QzJe (1/2)**
>
> We thank the reviewer for the constructive comments and positive feedback on our paper. Regarding the concerns of the reviewer, we provide the following responses.
>
>
>
> **W1, Q1, Q5**: In our initial RLVR-only experiments, we indeed considered a weighted combination $s = \alpha c + (1-\alpha)i$ and even designed a scheduler for $\alpha$ to reduce hyperparameter tuning effort. Our goal was to avoid introducing yet another sensitive hyperparameter for RL training, especially at larger scales (e.g., 7B models). Empirically, we found that the configuration with $\alpha = 0.5$, temperature $T = 5.0$, and window size $W = 10$ works robustly in the RLVR setting and transfers reasonably well to the hybrid setup. For this reason, we opted to fix $s = c + i$ when training the 7B model. Below we provide the ablation studies for the trade off hyperparameters. We also explain why not using a larger value for W in the appendix (Figure 13)
>
> | Model                      | SAT  | ScienceQA | MathVista (mini) | ChartQA | InfoVQA | MMMU |
> |---------------------------|------|-----------|------------------|---------|---------|------|
> | MoDoMoDo                  | 50.0 | 65.7      | 32.2             | 70.4    | 59.9    | 39.1 |
> | Ours ( α = 0.50, T = 5.0, W=10 )                  | 55.2 | 71.6 | 33.2 | 70.4 | 60.8 | 42.4 |
> | α = 0.50, T = 1.0, W=10   | 52.0 | 71.6      | 33.4             | 68.1    | 58.5    | 40.4 |
> | α = 0.75, T = 5.0, W=10   | 54.4 | 71.2      | 32.9             | 70.0    | 60.7    | 41.0 |
> | α = 0.25, T = 5.0, W=10   | 55.3 | 72.2      | 33.7             | 66.1    | 56.8    | 39.3 |
> | α = 0.50, T = 5.0, W=50   | 51.9 | 70.5      | 32.9             | 69.9    | 59.7    | 40.8 |
>
>
> We have also tested the unweighted combination on the illustrative example (Figure 5 in the paper) and on the NYUv2 experiment in the appendix, where our method still yields the best performance among loss-reweighting baselines. This supports the generality of our approach across different benchmarks. Finally, because our method relies on sliding-window statistics, developing a fully rigorous theory without strong simplifying assumptions is non-trivial within the rebuttal period. We are actively working on a more complete theoretical treatment and will update this thread as soon as we have any interesting results.
>
>
>
> **W2, Q2**: Thank you for the suggestion. We have updated the experimental setup in Section A.1 of the appendix, where we now include detailed dataset statistics and additional hyperparameters. We also clarify that we do not perform reward normalization: our method is magnitude-agnostic, as it operates on relative convergence and instability during training, which does not require further normalization. Due to the large scale of our experiments, we follow prior work and report the results from a single run for each configuration [1-5].
>
>
> **W3,Q3**: Thank you for your suggestion, we have added a limitation section after the main paper. Under a fixed training budget, it is expected that there will remain some categories where our method does not achieve the best performance, because some models are already close to a Pareto front for the corresponding tasks—further improving any one axis can hurt others. This trade-off is also routinely observed in the MTL literature (see Table 4 in the appendix), where the overall best method may lag behind certain alternatives on specific domains while substantially outperforming them on others.
>
> **W4**: For typical settings (e.g., Qwen-3B/7B, $K<10$ objectives, window size $W=10$ ), our method introduces only $\Theta(K W)$ extra scalar operations and $\Theta(K W)$ memory, which is negligible compared to the $\Theta(10^{11})-\Theta(10^{12})$ FLOPs per step of the underlying model; in practice we observed no measurable slowdown. Even on a much smaller setup (Figure 16), our method causes no observable overhead compared with the single task baseline.
>
>
> **I1**: As we mentioned in the general response, the apparent “collapse” in performance for some model families relative to their base models is intentionally shown to highlight that our approach preserves general capabilities during reasoning finetuning. We provide detailed numbers in the general response to make this point explicit: while MiMo-VL-7B-RL preserves the original base model performance quite consistently across benchmarks, other model families often suffer from substantial catastrophic forgetting of these core skills.
>
> **I2**: In our context, “domain” and “dataset” can be used interchangeably, as both refer to the data sources used for training.
>
> **I3**: Thank you for the suggestion. We now dedicate Section A.6 in the appendix to a comparison with other MTL methods and to additional experiments.

---

> ### Author Response · Authors · 2025-11-25
> **Response to Reviewer QzJe (2/2)**
>
> **Q4**: Thank you for the insightful suggestion. We find that reducing the KL regularization coefficient $\beta$ helps boost reasoning performance on several math benchmarks (e.g., MathVista, MathVision, MathVerse). This observation is consistent with recent work that fully removes this term in similar settings [6-8]
>
>
> | Variant         | LISA | MMMU-PRO|  AI2D  | MathVista | MathVision | MathVerse  | MMBench | VizWiz | OCRBenchv2  |
> |----------------|-------|-------|-------|-------|-------|-------|-------|-------|-------|
> | Base model     | 65.13 | 25.55 | 67.62 | 61.70 | 9.54  | 26.29 | 71.82 | 50.82 | 39.49 |
> | LwF    $\beta=0.01$        | 65.08 | 29.59 | 73.93 | 63.90 | 18.42 | 33.98 | 73.11 | 53.12 | 39.56 |
> | LwF $\beta=0.001$| 64.97 | 30.41 | 74.81 | 64.20 | 20.13 | 34.61 | 73.84 | 54.07 | 39.74 |
> | LwF $\beta=0.0001$| 64.82 | 30.79 | 75.68 | 64.13 | 21.56 | 35.72 | 74.71 | 55.12 | 39.21 |
> | Ours           | 67.24 | 34.15 | 78.21 | 66.70 | 25.11 | 40.83 | 78.52 | 61.97 | 39.72 |
>
> **Q6**: We have added Figure 12 to illustrate how the per-objective coefficients evolve later in training. From this figure, we can rank the focus induced by our method (from low to high) as: format rewards, IoU reward, next-token prediction (on the OCR task), and reasoning accuracy. This aligns with our intuition that our method progressively shifts emphasis toward harder and more unstable objectives.
>
> [1] https://arxiv.org/abs/2504.11468
>
> [2] https://arxiv.org/abs/2503.17352
>
> [3] https://arxiv.org/abs/2503.06749
>
> [4] https://arxiv.org/abs/2504.07615
>
> [5] https://arxiv.org/abs/2503.07536
>
> [6] https://arxiv.org/abs/2503.14476
>
> [7] https://arxiv.org/abs/2503.20783
>
> [8] https://www.datocms-assets.com/64837/1763662397-1763646865-olmo_3_technical_report-1.pdf

---

### Official Review · Reviewer_huLW · 2025-11-01

**Soundness:** 2
**Presentation:** 1
**Contribution:** 2
**Rating:** 4
**Confidence:** 4

**Summary:**

This paper mainly shows the empirical results for the performance degradation of LLM on general capabilities such as perception and robustness after post-training the model. The authors said those kind of phenomenon is related to the catastrophic forgetting, and the main objective is to prevent this problem. In the methods, to prevent the forgetting, replaying the data on retaining the general ability can solve this problem. However, due to the difference on convergence rate between replayed task and the main objective (e.g. reasoning), the training instability occurs. To tackle this problem, the authors suggest the re-weighting mechanism to make the training stable. In the experiment, the proposed method outperforms other baselines in various benchmark, and in the ablation study, the authors show the effectiveness of re-weighting scheme.

**Strengths:**

1. The problem statement and the solution is very simple. The authors pointed out well on the problems on the catastrophic forgetting after the post-training, and it is natural to use replay to elevate the performance on  general ability. Furthermore, the re-weighting techniques on securing the training stability is also effective.

**Weaknesses:**

1. The overall presentation is not well-organized. At the introduction section, the authors mainly pointed out the occurrence of catastrophic forgetting on post-training on the LLM. However, in the method section, they shifted the attention to the re-weighting scheme to tackle the training instability problem. I think this flow is not quite convincing and little bit confused. It would be better to specify more details on the motivation behind the re-weighting. Just showing the empirical results in Figure 3 (middle) may be not enough to support the claim.

2. The re-weighting scheme is quite heuristic. We cannot guarantee that this method can also solve the problem on much larger models such as 32B models. I think the proposed weighting scheme highly depends on the training behavior of 7B models.

3. I wonder the re-weighting scheme really gives major contribution on the training stabilization. First, the performance difference between "Coreset" and the proposed method is marginal, and there are also some cases that "Coreset" outperforms the proposed method. Second, the authors do not show any empirical results on the training stabilization when we use the re-weighting scheme. The results on the excellence of the accuracy reward compared to the baseline and the thinking length do not give a persuasive point of view on the effectiveness of the re-weighting scheme.

**Questions:**

1. What kind of dataset you used for the replay? It would be better to specify the datasets.

---

> ### Author Response · Authors · 2025-11-25
> **Response to Reviewer huLW**
>
> Thank you for the detailed and constructive feedback. We treasure the opportunity to address your concerns and improve our work.
>
> **W1.** We have reorganized parts of the manuscript and will continue to iterate on the presentation; please let us know if you have any additional suggestions. We have added Figure 2, which shows how the forgetting issue emerges during RLVR fine-tuning (taken from the Hybrid setup). This motivates replaying general-capability data (we also see that all replay-aided methods consistently preserve the general performance in this setting). However, doing so introduces heterogeneous objective types beyond those in the original RLVR pipeline, which in turn requires appropriate reweighting. For example, we observe two extremes: Coreset and Reasoning-only perform quite well on reasoning tasks but perform poorly on segmentation and OCR, whereas the Default excels at segmentation but underperforms on reasoning.
>
> **W2.** We would like to clarify that our reweighting mechanism is not model-dependent. In MoDoMoDo, multiple proxy models are trained to learn test performance as a function of the reweighting coefficients, and the resulting mapping is not guaranteed to transfer across different model scales. In contrast, our reweighting coefficients are adapted online based on the model’s training behavior: the convergence-rate and instability signals capture how well the current model is being optimized for a given objective, regardless of its size. This is similar in spirit to several MTL methods discussed in Section A.6 of the appendix, where we also benchmark our reweighting mechanism on a 2D illustrative example and on an additional scene-understanding experiment with a 150M model and effectively show its benefit.
>
> **W3.** (Please also refer to the general response) We would also like to clarify that our method is not primarily designed to stabilize training, but rather to adjust the relative strength of heterogeneous objectives so that no single objective dominates. Given a fixed training budget, it is therefore expected that there will be some benchmarks where our method does not achieve the best performance, especially when other models lie closer to the Pareto front for a particular task—further optimizing that axis can hurt other metrics. For instance, Coreset performs slightly better than our method on some reasoning-heavy tasks (e.g., AI2D and MathVista) because it heavily downweights general-capability data and thus biases training toward reasoning domains. This trade-off is consistent with observations in the MTL literature (Table 4 in the appendix), where the best overall method can lag behind some baselines on specific domains while being substantially better on others. In Figure 12, we show that near the end of training, the reweighting focuses most on the reasoning-accuracy reward and next-token prediction, while assigning less weight to formatting; together with Figure 4, we hope this clearly illustrates the effectiveness of our method compared to uniform baselines.
>
> **Q1.** We have added detailed statistics of all datasets used in our experiments, along with their associated rewards/objectives, in Section A.1 of the appendix.

---

### Author Response · Authors · 2025-11-25
**General Response to Reviewers**

We thank all reviewers for their careful and constructive evaluation of our work on investigating and mitigating catastrophic forgetting in reasoning models. In this general response, we are addressing a number of important points raised by the reviewers:
- **Stronger justification of our proposed reweighting mechanism and a comparison against established multi-task learning (MTL) techniques:** In response, we dedicate Section A.6 of the appendix to positioning our method among standard MTL approaches on two common benchmarks. There, we show that our reweighting mechanism is competitive and demonstrates good generality across heterogeneous task types and model scales.
- **Radar chart in Figure 1:** We acknowledge that some model curves overlap on certain benchmarks (e.g., MiMo-VL-7B-SFT and MiMo-VL-7B-RL). Yet, our intention was precisely to highlight that MiMo-VL-7B-RL slightly lags behind its base model thank to their undisclosed Mixed On-policy Reinforcement Learning strategy. To make this clearer, we provide the detailed comparison of each reasoning model along with their base model on 3 different model categories on below table.

||Model|A-OKVQA| AesBench | VStar  | VisOnly | OCRBench | R-Bench-Dis|Average|
|-|-|-|-|-|-|-|-|-|
| Base model |**MiMo-VL-7B-SFT**| 52.66  | 29.44 | 81.15 | 48.67  | 85.10   | 57.17 |   59.03 |
|| MiMo-VL-7B-RL| 55.59 (+2.93) | 29.17 (-0.28) | 79.06 (-2.09) | 42.67 (-6.00)  | 84.90  (-0.20)| 55.96   (-1.21)   |  57.89  (-1.14) |
||**Qwen2.5-VL-3B-Instruct** | 85.59  | 59.17| 75.39 | 42.22  | 82.60   | 62.22      |     67.87    |
|| VLAA-Thinker-Qwen2.5VL-3B | 83.76 (-1.83) | 51.94 (-7.22) | 58.12 (-17.28) | 35.56 (-6.67) | 75.20  (-7.40) | 59.19 (-3.03)     |  60.63 (-7.24)|
|| Qwen2.5VL-3b-RLCS| 38.34 (-47.25) | 40.28  (-18.89)     | 29.84 (-45.55) | 26.89 (-15.33)  | 80.60  (-2.00) | 38.79  (-23.43)    |   42.46 (-25.41)      |
|| SpaceThinker-Qwen2.5VL-3B      | 84.37 (-1.22)  | 60.00  (+0.83)     | 66.49 (-8.90) | 35.56 (-6.67)  | 80.80  (-1.80) | 61.41   (-0.81)   |   64.77 (-3.09)     |
| Base model | **Qwen2.5-VL-7B-Instruct**    | 88.73  | 57.22| 76.44 | 50.44  | 88.40   | 65.45 |71.12|
| | VLAA-Thinker-Qwen2.5VL-7B  | 86.38 (-2.36) | 55.00 (-2.22) | 60.73 (-15.71) | 49.56 (-0.89)  | 83.00  ( -5.40) | 64.44 (-1.01) | 66.52 (-4.60)     |
| | WeThink-Qwen2.5VL-7B | 79.48 (-9.26) | 49.44 (-7.78) | 70.16 (-6.28) | 44.22  (-6.22) | 88.70 (+0.30)  | 61.62  (-3.84) | 65.60  (-5.51)  |

- **Non-best performance on some tasks:** On some tasks (e.g., AI2D and MathVista), Coreset slightly outperforms our method. We believe this is because Coreset strongly downweights general-capability data, effectively biasing training toward reasoning-heavy domains. However, this comes at the cost of degraded general capabilities (e.g., a 4% drop on OCRBench-v2 and the lowest LISA score among replay-based baselines), whereas our method explicitly aims to preserve both reasoning and general skills by obtaining highest scores on those two benchmarks. To further close the gap on reasoning-centric tasks, we reduce the temperature from T=5.0 to T=1.0 to allow a more aggressive reweighting. As a result, this sharper prioritization reduces the relative emphasis on scene-understanding data and places more weight on reasoning objectives (Figures 3 and Figure 12), which in turn boosts reasoning performance and helps our method surpass Coreset on the aforementioned benchmarks.

| Variant         | LISA | MMMU-PRO|  AI2D  | MathVista | MathVision | MathVerse  | MMBench | VizWiz | OCRBenchv2  |
| - | - | -| -| - | - | -| -| -| -|
| Coreset  | 64.82 | 31.91 | 79.92 | 66.90 | 23.36 | 37.58 | 78.09 | 63.76 | 35.49 |
| Ours (T=5.0) | 67.24 | 34.15 | 78.21 | 66.70 | 25.11 | 40.83 | 78.52 | 61.97 | 39.72 |
| Ours (T=1.0) | 66.93 | 34.58 | 80.17 | 68.30 | 25.42 | 40.97 | 78.45 | 63.52 | 39.61 |

- **Justification of the default hyperparameter set**: Due to the large computational requirements of RL training, we find that setting $T = 5$ and $\alpha = 0.5$ works reasonably well in the RLVR-only setting. For simplicity, we keep this configuration for the Hybrid setup and do not perform additional hyperparameter tuning in the large-scale setting. We provide the results when varying the trade-off $\alpha$, the temperature T, and the window size W in individual responses. Although upweighting the instability term can increase the weight assigned to the accuracy reward since this term is highly fluctuating, it comes at the cost of sacrificing essential perception skills. For example, $\alpha = 0.25$ improves performance on SAT and ScienceQA by 0.1\% and 0.6\%, respectively, but reduces ChartQA and InfoVQA performance by 3\%. Similar to our illustrative example, decreasing the temperature induces higher variation across tasks—for instance, it yields the highest score on MathVista while reducing SAT performance by 3.2\%.

We address all remaining points raised by the reviewers in the individual responses below.

---

### Note · Authors · 2026-01-06

I have read and agree with the venue's withdrawal policy on behalf of myself and my co-authors.